# Metabolic disruption impairs ribosomal protein levels, resulting in enhanced aminoglycoside tolerance

Rauf Shiraliyev, Mehmet A Orman*

William A. Brookshire Department of Chemical and Biomolecular Engineering, University of Houston, Houston, United States

**Abstract** Aminoglycoside antibiotics target ribosomes and are effective against a wide range of bacteria. Here, we demonstrated that knockout strains related to energy metabolism in *Escherichia coli* showed increased tolerance to aminoglycosides during the mid-exponential growth phase. Contrary to expectations, these mutations did not reduce the proton motive force or aminoglycoside uptake, as there were no significant changes in metabolic indicators or intracellular gentamicin levels between wild-type and mutant strains. Our comprehensive proteomics analysis unveiled a noteworthy upregulation of proteins linked to the tricarboxylic acid (TCA) cycle in the mutant strains during the mid-exponential growth phase, suggesting that these strains compensate for the perturbation in their energy metabolism by increasing TCA cycle activity to maintain their membrane potential and ATP levels. Furthermore, our pathway enrichment analysis shed light on local network clusters displaying downregulation across all mutant strains, which were associated with both large and small ribosomal binding proteins, ribosome biogenesis, translation factor activity, and the biosynthesis of ribonucleoside monophosphates. These findings offer a plausible explanation for the observed tolerance of aminoglycosides in the mutant strains. Altogether, this research provides valuable insights into the mechanisms of aminoglycoside tolerance, paving the way for novel strategies to combat such cells.

*For correspondence:
morman@central.uh.edu

Competing interest: The authors declare that no competing interests exist.

## eLife assessment

The findings of this study are **valuable** as they challenge the dogma regarding the link between lowered bacterial metabolism and tolerance to aminoglycosides. The authors propose that the well-known tolerance to AG of mutants such as those of complexes I and II is not due to a decrease in the proton motive force and thus antibiotic uptake. The results presented here are **convincing**.

## Introduction

Global public health is currently confronted with a pressing challenge in the form of antimicrobial resistance (*Binder et al., 1999*). Addressing this issue necessitates a comprehensive understanding of the diverse mechanisms used by bacterial cells to survive antibiotic treatments. These mechanisms can be broadly classified into two groups: reversible and irreversible. Reversible mechanisms, often termed 'tolerance' mechanisms, are not genetically inherited and encompass alterations in bacterial growth and behavior. Within this category, antibiotic-tolerant cells, such as persisters, viable but nonculturable cells, and stationary-phase cells, transiently survive high antibiotic concentrations. Their formation is linked to stochastic and/or deterministic processes triggered by various stress factors including the SOS response, stringent response, reactive oxygen species, nutrient depletion, and overpopulation (*Amato et al., 2013*; *Germain et al., 2015*; *Helaine and Kugelberg, 2014*; *Molina-Quiroz*

**eLife digest** Bacteria that are resistant to antibiotic drugs pose a significant challenge to human health around the globe. They have acquired genetic mutations that allow them to survive and grow in the presence of one or more antibiotics, making it harder for clinicians to eliminate such bacteria from human patients with life-threatening infections.

Some bacteria may be able to temporarily develop tolerance to an antibiotic by altering how they grow and behave, without acquiring any new genetic mutations. Such drug-tolerant bacteria are more likely to survive long enough to gain mutations that may promote drug resistance.

Recent studies suggest that genes involved in processes collectively known as energy metabolism, which convert food sources into the chemical energy cells need to survive and grow, may play a role in both tolerance and resistance. For example, *Escherichia coli* bacteria develop mutations in energy metabolism genes when exposed to members of a family of antibiotics known as the aminoglycosides. However, it remains unclear what exact role energy metabolism plays in antibiotic tolerance.

To address this question, Shiraliyev and Orman studied how a range of *E. coli* strains with different genetic mutations affecting energy metabolism could survive in the presence of aminoglycosides. The experiments found that most of the mutant strains had a higher tolerance to the drugs than normal *E. coli*. Unexpectedly, this increased tolerance did not appear to be due to the drugs entering the mutant bacterium cells less than they enter normal cells (a common strategy of drug resistance and tolerance).

Further experiments using a technique, known as proteomics, revealed that many genes involved in energy metabolism were upregulated in the mutant bacteria, suggesting these cells were compensating for the genetic abnormalities they have. Furthermore, the mutant bacteria had lower levels of the molecules the antibiotics target than normal bacteria.

The findings of Shiraliyev and Orman offer critical insights into how bacteria become tolerant of aminoglycoside antibiotics. In the future, this may guide the development of new strategies to combat bacterial diseases.

---

*et al., 2018*; *Theodore et al., 2013*; *Van den Bergh et al., 2017*; *Wood et al., 2013*). The reversible nature of these cells allows them to switch between antibiotic-tolerant and sensitive states. On the other hand, irreversible mechanisms, often termed 'resistance' mechanisms, are heritable and related to mutagenic processes (*Balaban et al., 2019*). Antibiotic-resistant mutants, falling into this category, can emerge due to mutations in antibiotic target proteins or proteins that contribute to enhanced repair mechanisms, cell dormancy, drug efflux systems, and alternative mechanisms that circumvent the antibiotic target (*C Reygaert, 2018*; *Darby et al., 2023*; *Lambert, 2005*; *Munita and Arias, 2016*; *Okusu et al., 1996*).

Aminoglycosides were among the earliest antibiotics used in clinical practice (*Becker and Cooper, 2013*; *Krause et al., 2016*). These antibiotics are a class of naturally occurring or semisynthetic amino-modified sugars known for their broad-spectrum activity against diverse bacterial species (*Aggen et al., 2010*; *Endimiani et al., 2009*; *Ikäheimo et al., 2000*; *Karlowsky et al., 2003*; *Landman et al., 2010*; *Ristuccia and Cunha, 1985*; *Sader et al., 2015*). Since the discovery of the first aminoglycoside antibiotic, streptomycin, which was isolated from *Streptomyces griseus* (*Kresge et al., 2004*; *Schatz et al., 2005*; *Woodruff, 2014*), numerous other aminoglycosides have been identified, including gentamicin, kanamycin, tobramycin, amikacin, and neomycin (*Becker and Cooper, 2013*; *Krause et al., 2016*). Although the use of aminoglycosides declined for a brief period with the advent of newer antibiotics (such as fluoroquinolones) which were thought to have lower toxicity (*Becker and Cooper, 2013*; *Krause et al., 2016*), the rise of resistance to these new drugs has revived interest in aminoglycosides and the development of new ones with improved dosing schemes (*Becker and Cooper, 2013*; *Krause et al., 2016*).

Studies on the mechanism of action of aminoglycosides against bacteria have shown that they disrupt protein synthesis by targeting the ribosome. The primary action of aminoglycosides is to bind to the 16S ribosomal RNA of bacteria, a component of the 30S ribosomal subunit (*Moazed and Noller, 1987*; *Recht et al., 1999*; *Woodcock et al., 1991*). The initial entry of antibiotics into cells causes misreading in protein synthesis as the antibiotics interact with the ribosomes involved in

chain elongation. Then, these misread proteins or polypeptides are incorporated into the membrane, causing membrane damage (*Davis et al., 1986*). This increases the amount of antibiotics entering the cells, leading to more misreading, and the formation of more channels, and eventually the complete inhibition of protein synthesis (*Davis et al., 1986*). Moreover, aminoglycosides potentially inhibit ribosome recycling by binding to RNA helix 69 (H69) of the 50S ribosomal subunit, which can also lead to the inhibition of mRNA and tRNA translocation (*Borovinskaya et al., 2007*).

It has been demonstrated that antibiotic-tolerant cells can become susceptible to aminoglycosides by metabolizing certain carbon sources (*Allison et al., 2011*). This susceptibility arises from an enhanced drug uptake as a result of an increase in the electron transport chain (ETC) activity and membrane potential, facilitated by the breakdown of these specific carbon sources (*Allison et al., 2011*). The process of aminoglycoside uptake is a unique, energy-requiring mechanism where the electrochemical potential across the cytoplasmic membrane and electron flow through membrane-bound respiratory chains are believed to be significant factors (*Taber et al., 1987*). However, the bactericidal effect of aminoglycosides may not result from the downstream impacts of voltage-dependent drug uptake, but rather from an irregular membrane potential (*Bruni and Kralj, 2020*). *Bruni and Kralj, 2020* suggested that hyperpolarization, stemming from changes in ATP flux due to the reversal of F1Fo-ATPase activity, could potentially intensify aminoglycoside-mediated cell death.

Previous studies underscore the potential role of metabolic mutations in aminoglycoside tolerance and resistance, a phenomenon that warrants further investigation (*Muir et al., 1981*; *Shan et al., 2015*). A recent study, which analyzed genomic alterations in *Escherichia coli* strains, including uropathogenic UTI89 strains, following daily antibiotic exposure (*Van den Bergh et al., 2022*), showed that mutations were predominantly detected in genes of the *nuo* operon, a vital component of bacterial energy metabolism. This highlights a potential link between metabolic adaptations and antibiotic tolerance. Additionally, research led by Collin's group uncovered genes related to central metabolism that contribute to antibiotic resistance in *E. coli* cells exposed to various antibiotics, including aminoglycosides (*Lopatkin et al., 2021*). These findings are corroborated by similar mutations identified in clinical *E. coli* pathogens, as evidenced by the examination of a comprehensive library of 7243 *E. coli* genomes from NCBI Pathogen Detection (*Lopatkin et al., 2021*). In our study, we found that knockout strains with genes related to the tricarboxylic acid (TCA) cycle and the ETC displayed increased tolerance to aminoglycosides. Intriguingly, this increased tolerance was not attributed to reduced proton motive force (PMF), which affects drug uptake, as evidenced by insignificant alterations in ATP levels or membrane potential in the mutant strains compared to the wild-type strain. We employed untargeted mass spectrometry to quantify proteins in the mutant and wild-type strains, revealing a notable upregulation of proteins associated with the TCA cycle in the mutants. This suggests that these strains compensate for the disruption in their energy metabolism by altering TCA cycle activity to maintain their membrane potential and ATP levels. Moreover, our pathway enrichment analysis underlined local network clusters that were consistently downregulated across all mutant strains. These clusters were related to both large and small ribosomal binding proteins, ribosome biogenesis, translation factor activity, and the biosynthesis of ribonucleoside monophosphates. These findings provide a credible rationale for the observed tolerance to aminoglycosides in the mutant strains.

## Results

### Deletions of the TCA cycle and ETC genes increased tolerance to aminoglycosides

Given the crucial role of energy metabolism in aminoglycoside tolerance, our initial objective was to assess various knockout strains that involved the deletion of genes associated with the TCA cycle, such as *sucA*, *gltA*, *mdh*, *sdhC*, *icd*, *acnB*, and *fumA*, as well as the NADH (nicotinamide adenine dinucleotide) dehydrogenase enzyme of ETC, including *nuoM* and *nuoI* from *E. coli* MG1655. To conduct the experiments, both wild-type and mutant strains were cultured overnight and then diluted 100-fold in fresh 2 ml Lysogeny Broth (LB) in test tubes. The cultures were grown until the mid-exponential phase (*t* = 3.5 hr) in a shaker at 37°C and 250 rpm. Subsequently, the cultures were exposed to various aminoglycosides (50 μg/ml streptomycin, 50 μg/ml gentamicin, and 50 μg/ml amikacin) for a duration of 5 hr. Samples were collected before and after the treatments and plated on LB agar to quantify the surviving cell fractions. Analysis of the results indicates that most of the knockout strains (Δ*sucA*,

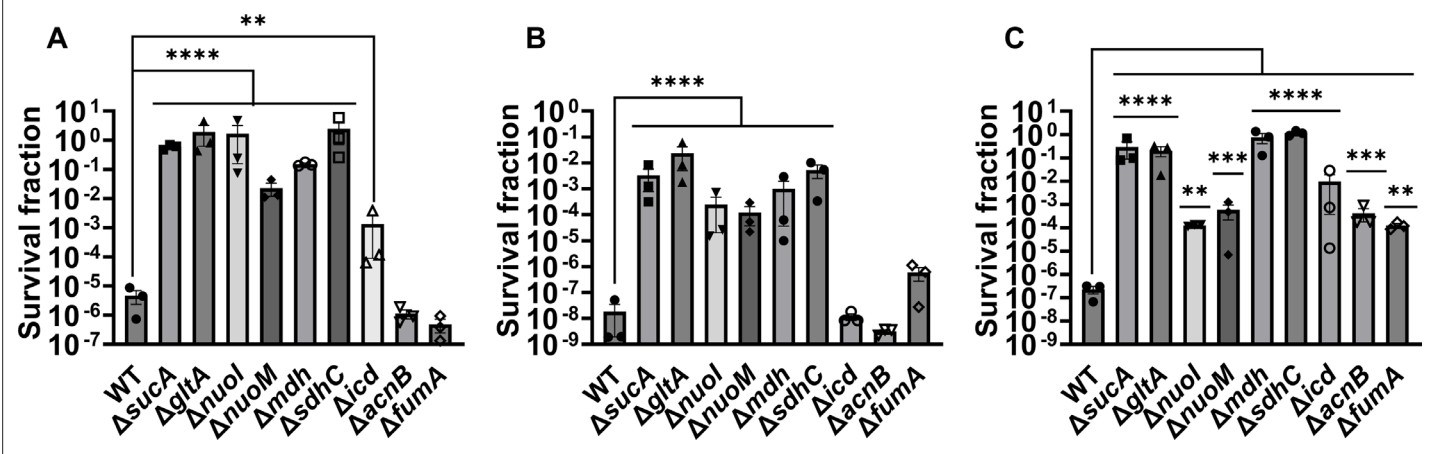

**Figure 1.** Deletions of the tricarboxylic acid (TCA) cycle and electron transport chain (ETC) genes enhanced tolerance to aminoglycosides. In the mid-exponential phase ($t = 3.5$ hr), *E. coli* MG1655 wild-type and knockout strains underwent (**A**) streptomycin, (**B**) gentamicin, and (**C**) amikacin treatments at a concentration of 50 µg/ml for a duration of 5 hr in test tubes. Following the treatments, cells were washed to eliminate the antibiotics and then plated on LB agar plates to quantify the surviving cell fractions. CFU: colony-forming units; WT: wild type. For pairwise comparisons, one-way analysis of variance (ANOVA) with Dunnett's post hoc test was used where **$p < 0.01$, ***$p < 0.001$, and ****$p < 0.0001$. $N = 3$. Data points represent mean and standard error.

The online version of this article includes the following figure supplement(s) for figure 1:

**Figure supplement 1.** Impact of lower gentamicin concentration on the tolerance of the mutant strains.

Δ*gltA*, Δ*mdh*, Δ*sdhC*, Δ*nuoM*, and Δ*nuoI*) exhibited increased tolerance to streptomycin, gentamicin, and amikacin treatments when compared to the wild-type strain (**Figure 1A–C**). However, it is worth noting that mutant strains such as Δ*icd*, Δ*acnB*, and Δ*fumA* did not always exhibit increased tolerance (**Figure 1A–C**), underscoring the complex interplay between energy metabolism and specific antibiotic tolerance, as well as the pleiotropic effects of the gene deletions. Moreover, the surviving cells measured in these assays may not necessarily represent antibiotic-resistant cells, as the antibiotic tolerance assays were conducted at concentrations well above the minimum inhibitory concentration (MIC) levels. The MIC levels of the strains for the tested antibiotics displayed no drastic alterations compared to those of the wild type, despite some minor variations among them (**Supplementary file 1a**).

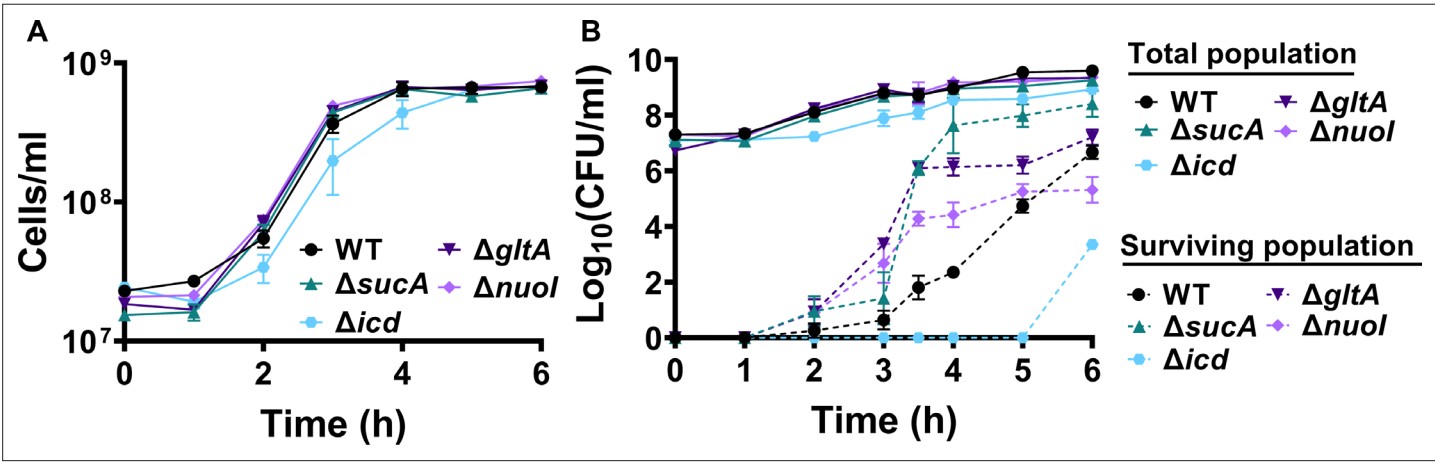

**Figure 2.** The observed tolerance in the mutant strains is not linked to cell growth. (**A**) Growth of *E. coli* MG1655 wild type, Δ*sucA*, Δ*gltA*, Δ*nuoI*, and Δ*icd* strains was assessed by measuring the number of cells per ml with flow cytometry. (**B**) The cells of both the *E. coli* MG1655 WT and mutant strains were collected from culture flasks (see Materials and methods for details) at indicated time intervals and then subjected to gentamicin treatment. The figure shows the colony-forming unit (CFU) levels of both the treated and untreated cultures, indicating the surviving and total cell populations, respectively. $N = 3$. Data points represent mean and standard error.

To gain a thorough understanding of the observed tolerance, we chose four mutant strains that exhibit either high tolerance (*sucA*, *gltA*, and *nuoI*) or low tolerance (*icd*) to gentamicin for the subsequent assays. Our initial objective was to delineate the time-dependent profiles of antibiotic-tolerant cells within cultures. To achieve this, cells from overnight cultures were diluted in fresh media and cultured for 6 hr in LB in flasks (see Materials and methods). Hourly samples were collected for cell quantification and antibiotic tolerance assays. We opted for flow cytometry for precise cell quantification, a more reliable method than optical density measurements (*Mohiuddin et al., 2020*; *Figure 2A*). In the antibiotic tolerance assays, samples were treated with gentamicin (50 µg/ml) for 5 hr and plated before and after the treatments (*Figure 2B*). Notably, the observed tolerance in these mutant strains is not linked to cell growth. For instance, the mutant strains associated with *sucA*, *gltA*, and *nuoI* genes exhibited higher tolerance to aminoglycosides although they did not show significantly altered cell growth compared to the wild type (*Figure 2A, B*). These mutant strains displayed a substantial increase in the number of tolerant cells (more than $10^4$-fold) between time points $t = 3$ and $t = 4$ hr during the mid-exponential growth phase when compared to the wild-type strain (*Figure 2B*). Conversely, while the *icd* mutant exhibited a reduction in cell growth, it was observed to be as sensitive to gentamicin as the wild type during the exponential growth phase and the formation of tolerant cells in this mutant strain was only evident after the time point $t = 5$ hr (*Figure 2A, B*). We emphasize that the observed tolerance in the mutant strains is transient or reversible, as they all exhibit high sensitivity to aminoglycosides, similar to the wild type, during the lag phase of growth (*Figure 2B*). Given that antibiotic concentrations may influence the number of surviving cells, we also treated cells with 5 µg/ml gentamicin at the mid-exponential phase. Although we observed a greater number of surviving cells in all strains when lower gentamicin concentrations were employed compared to higher concentrations, we still observed a similar trend; ΔsucA, ΔgltA, and ΔnuoI exhibited higher levels of surviving cells compared to Δicd and the wild-type strain (*Figure 1—figure supplement 1*).

## Energy-dependent aminoglycoside uptake is not a contributing factor

Aminoglycoside uptake is a unique and energy-requiring mechanism that depends on the electrochemical potential across the cytoplasmic membrane (*Taber et al., 1987*). Initially, we hypothesized that the genetic perturbations in these strains may have decreased PMF and aminoglycoside uptake. The mutant strains exhibiting increased aminoglycoside tolerance demonstrated no consistent pattern in metabolic activities during the mid-exponential growth phase, which was assessed using redox sensor green (RSG) dye for the indicated time points ($t = 3, 4, 5,$ and $6$ hr) of the cell growth (*Figure 3A*). When RSG molecules are reduced by bacterial reductases, critical components of energy metabolism, these molecules emit green fluorescence. This fluorescence signal should be suppressed by the presence of a metabolic inhibitor, such as carbonyl cyanide 3-chlorophenylhydrazone (CCCP) (*Figure 3—figure supplement 1*). Given that ATP is a pivotal product of PMF, we also quantified intracellular ATP levels in both wildtype and mutant strains, employing the BacTiter-Glo Microbial Cell Viability Assay (Catalog# G8230, Promega Corporation, Madison WI). This assay utilizes a single reagent to lyse cells and produce luminescence through the luciferase reaction, with the luminescent signal being directly proportional to the ATP content (*Figure 3—figure supplement 2*). We found no consistent pattern in ATP levels between the antibiotic-sensitive and tolerant strains, particularly during the mid-exponential growth phase (at $t = 3$ and $4$ hr) (*Figure 3B*). While the ΔsucA mutant generally showed reduced ATP levels, this was not the case for the other strains, which showed increased ATP levels around $t = 6$ hr (*Figure 3B*).

Furthermore, since the proton gradient is a vital element of PMF and a reduced $H^+$ ion gradient across the cell membrane is linked to reduced membrane potential, we conducted measurements of intracellular pH using the ratiometric pHluorin, known as a pH-sensitive derivative of green fluorescent protein (GFP) (*Miesenböck et al., 1998*). This GFP variant exhibits a bimodal excitation spectrum characterized by peaks at 410 and 470 nm, along with an emission maximum at 530 nm (*Martinez et al., 2012*). When subjected to acidification, the excitation at 410 nm diminishes while the excitation at 470 nm concurrently rises, which allows us to construct standard curves for the measurement of intracellular pH (*Figure 3—figure supplement 3*). For these assessments, we introduced pGFPR01 plasmids, where pHluorin is expressed under the arabinose-induced promoter, into the mutant strains utilized. Our results revealed that there was no anticipated acidification of the cytoplasm in the TCA

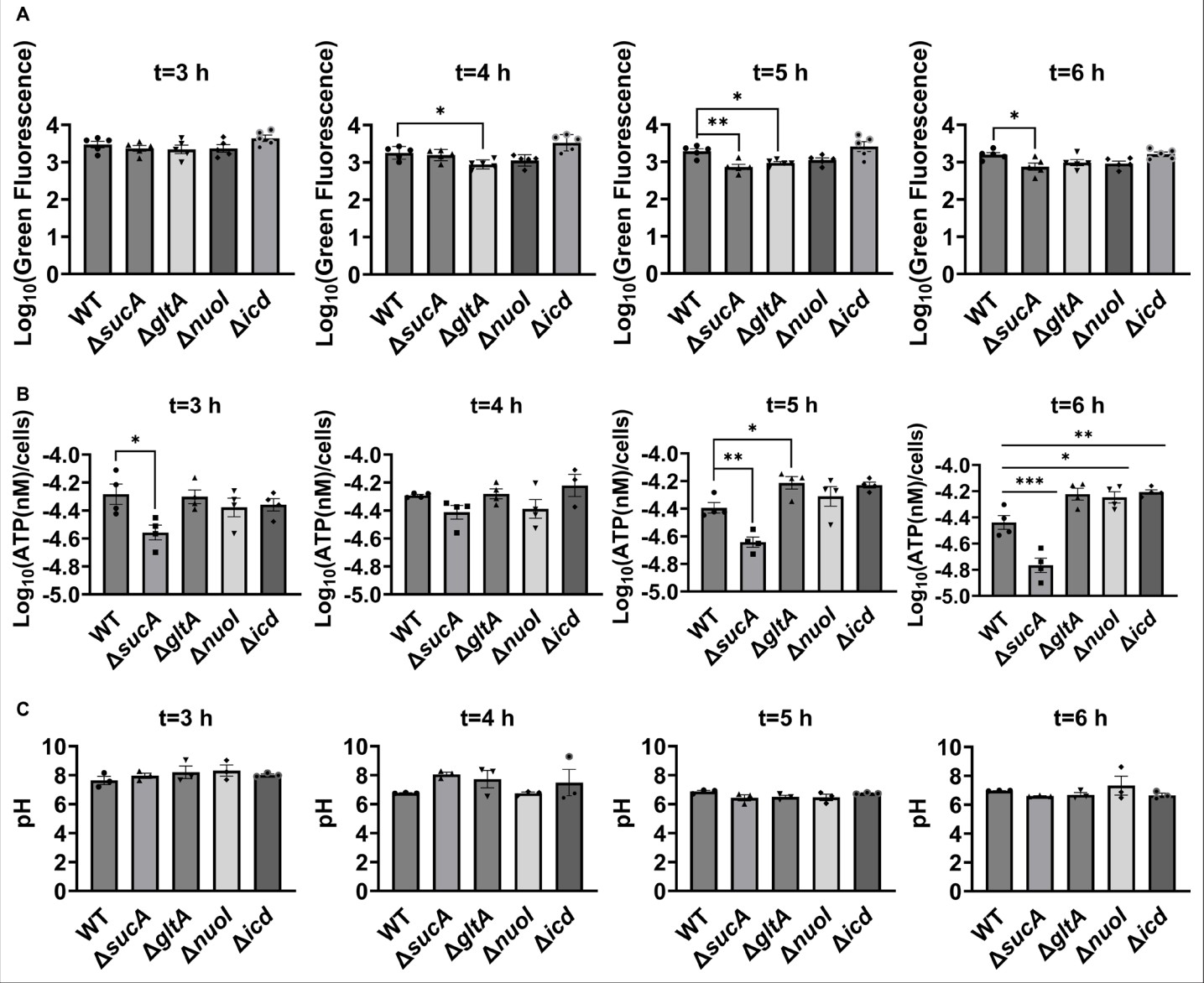

**Figure 3.** Comparable ATP levels, redox activities, and cytoplasmic pH levels were observed between the wild-type and mutant strains. (**A**) Redox sensor green (RSG) staining was conducted by suspending wild-type and mutant strains in 0.85% sodium chloride solution during the mid-exponential and early stationary phases, as outlined in the Materials and methods section. (**B**) The ATP levels were measured in wild-type and mutant cells during the mid-exponential and early stationary growth phases. (**C**) A comparison of the cytoplasmic pH of the WT and mutant cells during the mid-exponential and early stationary phases was performed using the ratiometric pHluorin. For pairwise comparisons, one-way analysis of variance (ANOVA) with Dunnett's post hoc test was used where *p < 0.05, **p < 0.01, and ***p < 0.001. N ≥ 3. Data points represent mean and standard error.

The online version of this article includes the following figure supplement(s) for figure 3:

**Figure supplement 1.** Control staining for bacterial redox activities.

**Figure supplement 2.** The ATP standard curve.

**Figure supplement 3.** The pH standard curve.

and ETC mutants compared to the wild type, both during the mid-exponential and early stationary phases (*Figure 3C*).

Finally, we utilized fluorophore-labeled aminoglycoside (Gentamicin-Texas Red, or GTTR) to investigate the cellular uptake of the drug. Specifically, cells in the mid-exponential phase of both wild-type and mutant strains (at *t* = 3.5 hr when a significant increase in gentamicin tolerance was observed in the mutant strains) were exposed to GTTR for an hour, followed by the analysis of cells using flow

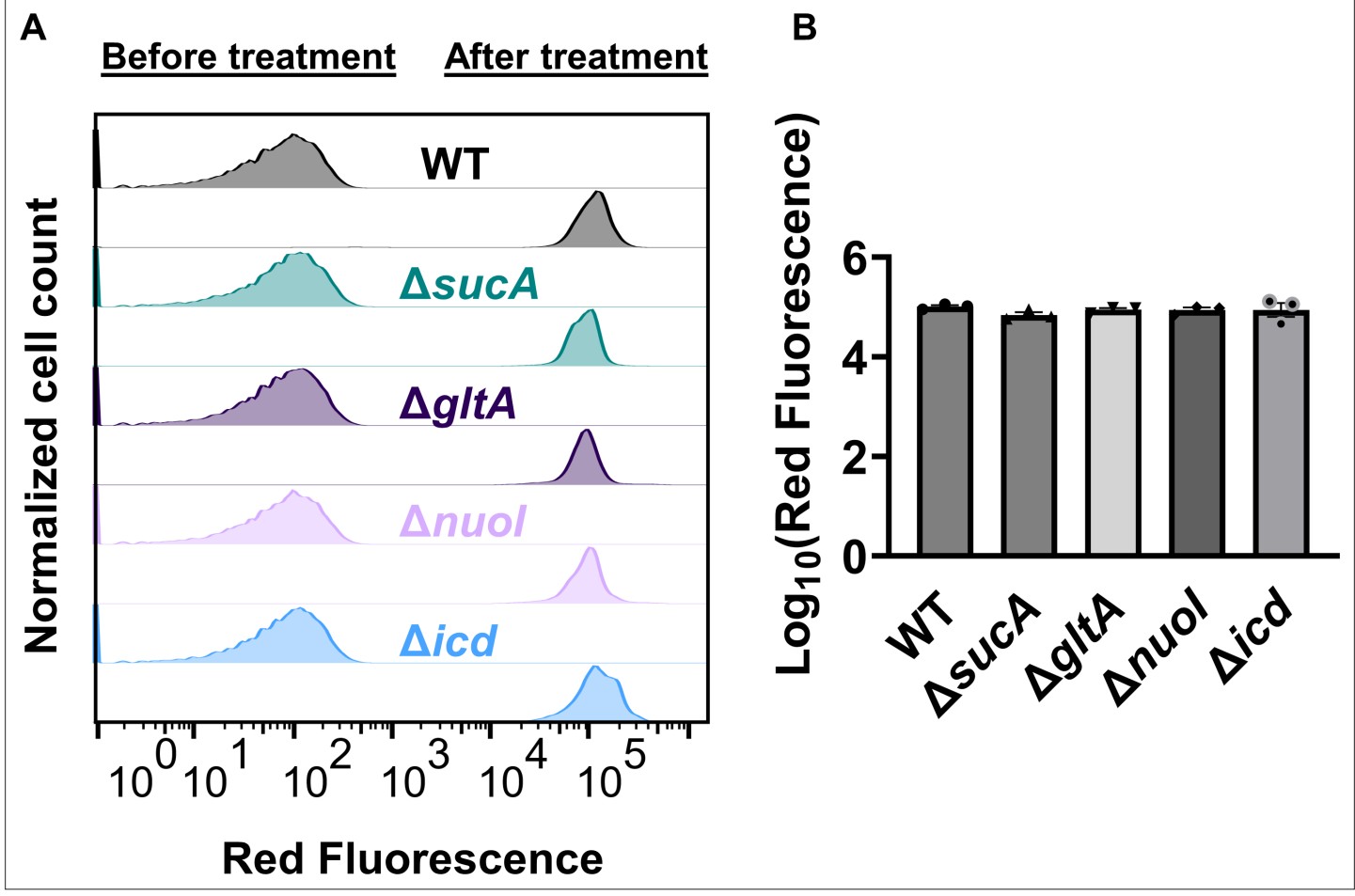

**Figure 4.** Deletion of genes associated with the tricarboxylic acid (TCA) cycle and electron transport chain (ETC) did not alter the drug uptake.
(**A**) Exemplary quantification of Gentamicin-Texas Red (GTTR) uptake in cells during the exponential growth phase ($t$ = 3.5 hr). (**B**) GTTR assays were performed on cells in the mid-exponential phase for both the wild-type and mutant strains, followed by fluorescence measurement using flow cytometry after one hour. The concentration of GTTR is 25 µg/ml. For pairwise comparisons, one-way analysis of variance (ANOVA) with Dunnett's post hoc test was used (no statistical significance was detected). $N$ = 3. Data points represent mean and standard error.

The online version of this article includes the following figure supplement(s) for figure 4:

**Figure supplement 1.** Deletion of genes associated with the tricarboxylic acid (TCA) cycle and electron transport chain (ETC) did not significantly alter the drug uptake.

cytometry. A short-term treatment was preferred, in line with a previous study (**Bruni and Kralj, 2020**) as cells treated with aminoglycosides become permeabilized at later time points (**Bruni and Kralj, 2020**), potentially introducing artificial impacts on drug uptake. Notably, the deletion of genes related to the TCA cycle and the ETC did not induce a significant alteration in GTTR uptake in cells during the exponential phase when compared to the wild type (**Figure 4A, B**). We note that the aminoglycoside concentration utilized in this study exceeds that of previous studies demonstrating the correlation between aminoglycoside uptake and PMF (**Taber et al., 1987**). Given that higher concentrations of aminoglycosides (>30 µg/ml) might obscure energy-dependent aminoglycoside uptake (**Taber et al., 1987**), we investigated a lower concentration of GTTR (5 µg/ml) and observed similar GTTR uptake across all tested strains (**Figure 4—figure supplement 1**). Collectively, these findings from multiple approaches strongly suggest that energy-dependent drug uptake is not the primary determinant of the observed antibiotic tolerance.

## Membrane potential dysregulation is not associated with the observed aminoglycoside tolerance

Fluorescent sensors for voltage and calcium have been utilized to monitor electrophysiology in bacteria at the single-cell level (*Bruni and Kralj, 2020*), and the findings revealed that the dysregulated membrane voltage was not essential for aminoglycoside uptake or inner membrane pore formation in *E. coli*, but it proved crucial for bactericidal activity (*Bruni and Kralj, 2020*). To assess the significance of dysregulated PMF, we employed a well-established assay based on 3,3'-dipropylthiadicarbocyanine iodide [DiSC$_3$(5)] (*Stokes et al., 2020*; *Wu et al., 1999*), a fluorescent dye commonly used for monitoring cell membrane potential. During cell hyperpolarization, DiSC$_3$(5) infiltrates the cell membrane,

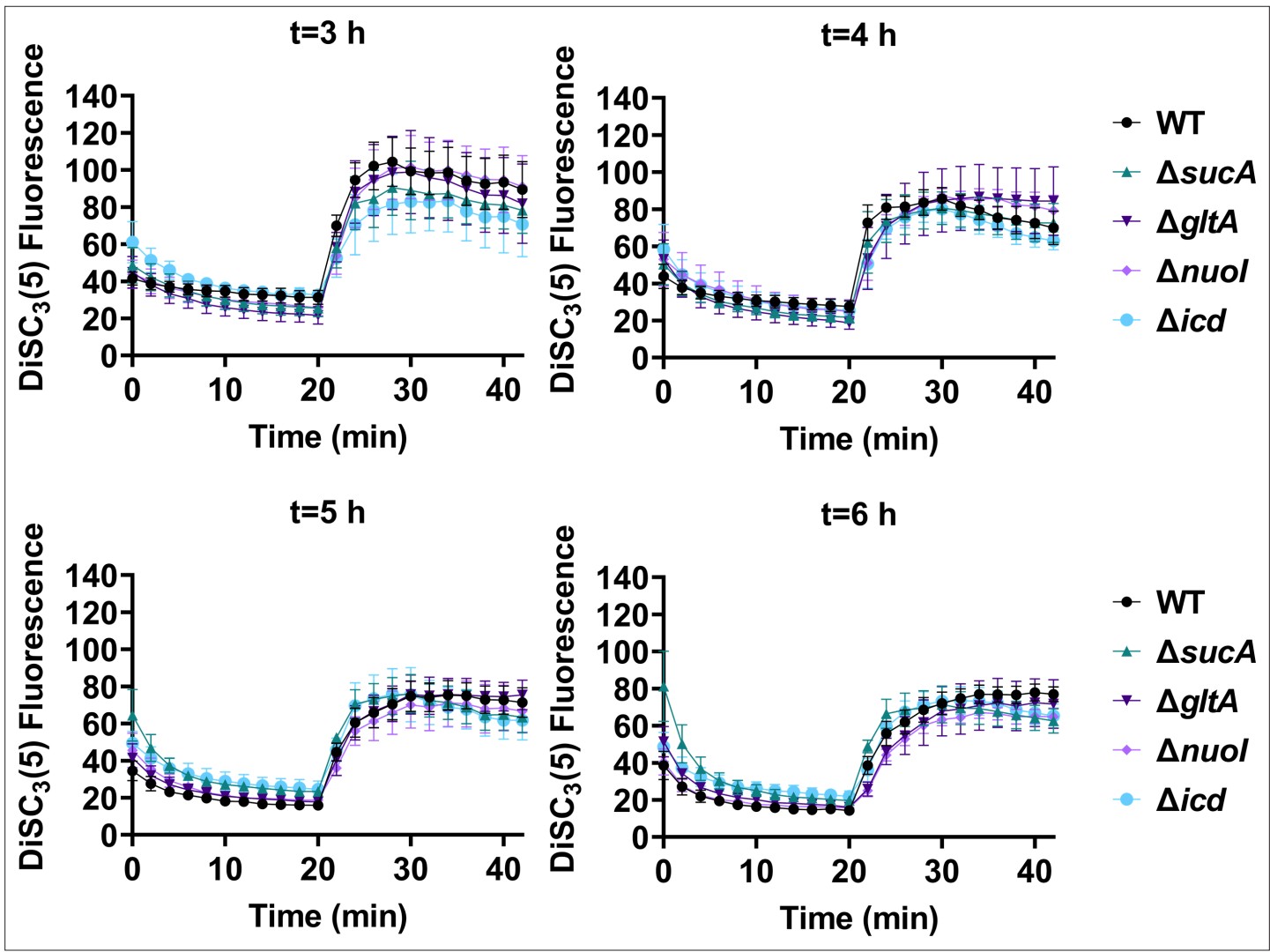

**Figure 5.** The dysregulation of membrane potential is not correlated with the observed aminoglycoside tolerance. Deletions of the tricarboxylic acid (TCA) cycle and electron transport chain genes resulted in no significant change in proton motive force (PMF). Mid-exponential and early stationary phase cells of wild-type and mutant strains were stained with DiSC$_3$(5), and at specified time intervals their fluorescence was measured using a plate reader. Cells were treated with gentamicin (50 µg/ml) after 20 min and fluorescence was measured again. $N$ = 4. Data points represent mean and standard error.

The online version of this article includes the following figure supplement(s) for figure 5:

**Figure supplement 1.** Polymyxin B, functioning as a control, leads to the disruption of the cytoplasmic membrane and proton motive force (PMF).

**Figure supplement 2.** The dysregulation of proton motive force (PMF) is not correlated with the observed aminoglycoside tolerance.

**Figure supplement 3.** Deletions of the ATP synthase genes did not confer a reverse ATPase-mediated killing under the conditions studied here.

**Figure supplement 4.** Deletions in the ATP synthase genes did not confer a reverse ATPase-mediated killing under the conditions studied here.

undergoing self-quenching and resulting in reduced fluorescence intensity. Conversely, during depolarization, the fluorescence intensity of DiSC$_3$(5) increases as it exits the cell membrane. Polymyxin B served as a control (*Figure 5—figure supplement 1*), as the polycationic peptide ring of polymyxin B binds to a negatively charged site within the lipopolysaccharide layer of the cellular membrane (*Domingues et al., 2012*), leading to the dissipation of the electric potential gradient which enhances the fluorescence intensity of DiSC$_3$(5) (*Figure 5—figure supplement 1*). To assess the effect of gentamicin on membrane potential, metabolically active cells from exponential phase cultures at time points $t$ = 3, 4, 5, and 6 hr were transferred to a buffer solution containing 1 µM DiSC$_3$(5) (*Figure 5*). Upon reaching equilibrium, the cells were treated with gentamicin. At designated time points, samples were collected, and fluorescence levels were measured using a plate reader (*Figure 5*). Similar to polymyxin B treatments, gentamicin disrupts the electric potential gradient of PMF, leading to an increase in the fluorescence intensity of DiSC$_3$(5) (*Figure 5*). Despite significant and rapid alterations in membrane potential observed upon gentamicin treatment, there was no significant difference in membrane potential between aminoglycoside-sensitive and tolerant strains (*Figure 5*). Moreover, if any of the gene deletions had an impact on membrane potential, we would have expected to observe altered fluorescence intensity in the specific knockout strain compared to that of the wild type during the equilibrium stage (before the addition of the drug), as previously reported (*Mohiuddin et al., 2022*). However, no significant difference in DiSC$_3$(5) fluorescence intensities was observed among the strains tested during the equilibrium phase (the initial 20 min, as illustrated in *Figure 5*). Furthermore, we did not observe significant differences in PMF between mutant and wild-type strains when lower concentrations of gentamicin (5 µg/ml) were used (*Figure 5—figure supplement 2*).

The membrane potential dysregulation induced by aminoglycosides was previously attributed to the combined activity of NADH dehydrogenase and a reversed F1Fo-ATPase (*Bruni and Kralj, 2020*), and the knockout strains targeting the proton-conducting Fo domain (Δ*atpB*, Δ*atpE*, Δ*atpF*), as well as Δ*atpG*, exhibited increased colony-forming units in response to aminoglycoside treatments in *E. coli* BW25113 (*Bruni and Kralj, 2020*). However, when we tested ATP synthase knockout strains of *E. coli* BW25113 from the Keio collection, we did not observe a comparable trend as reported earlier (*Bruni and Kralj, 2020*). In fact, the strains Δ*atpB*, Δ*atpE*, Δ*atpF*, and Δ*atpG* exhibited similar sensitivity to gentamicin compared to the wild type under the conditions studied here (*Figure 5—figure supplement 3A*). Interestingly, we observed a drastic increase in gentamicin tolerance in the Δ*atpC* mutant strain (*Figure 5—figure supplement 3A*), which was not reported in the previous study (*Bruni and Kralj, 2020*), possibly due to the different experimental conditions used here. The *atpC* gene encodes an F1 complex subunit of ATP synthase, promoting motor activity in the direction of ATP production rather than the reversed direction (*Bruni and Kralj, 2020*; *Guo et al., 2019*). We acknowledge potential variations between *E. coli* BW25113 and *E. coli* MG1655 strains, due to differences in their genomic DNA. To address this, we deleted two Fo components (Δ*atpA* and Δ*atpB*) and two F1 components (Δ*atpC* and Δ*atpD*) from *E. coli* MG1655 and assessed their gentamicin survival profiles. While a moderate increase in tolerance was observed for mutant strains of F1 components during the mid-exponential phase ($t$ = 3 and 4 hr) compared to the wild type, no clear trend was observed for mutant strains of Fo components (*Figure 5—figure supplement 3B*). The upregulation of tolerance in one of the mutant strains of F1 components (Δ*atpD*) was also evident during the mid-exponential phase when cells were exposed to a lower concentration of gentamicin (*Figure 5—figure supplement 4*). Overall, despite some variations in tolerance among ATP synthase mutants across different *E. coli* strains (MG1655 vs. BW25113) and gentamicin concentrations (*Figure 5—figure supplement 3* and *Figure 5—figure supplement 4*), the observed gentamicin-induced dysregulation in membrane potential may not be the primary factor contributing to the differences in antibiotic tolerance levels between the wild-type and TCA cycle mutant strains studied here.

## Proteomic analysis reveals molecular responses in mutant strains, unveiling potential mechanisms underlying aminoglycoside tolerance

In order to gain further insights beyond the conducted experiments and elucidate the mechanism responsible for the observed tolerance in TCA cycle and ETC mutants, we utilized untargeted mass spectrometry to quantify proteins within these mutants and subsequently compared them to their wild-type counterparts. Proteomics data analysis, specifically involving the determination of protein fold change and calculation of p-value (using $F$-test and $t$-test), was carried out through a process involving

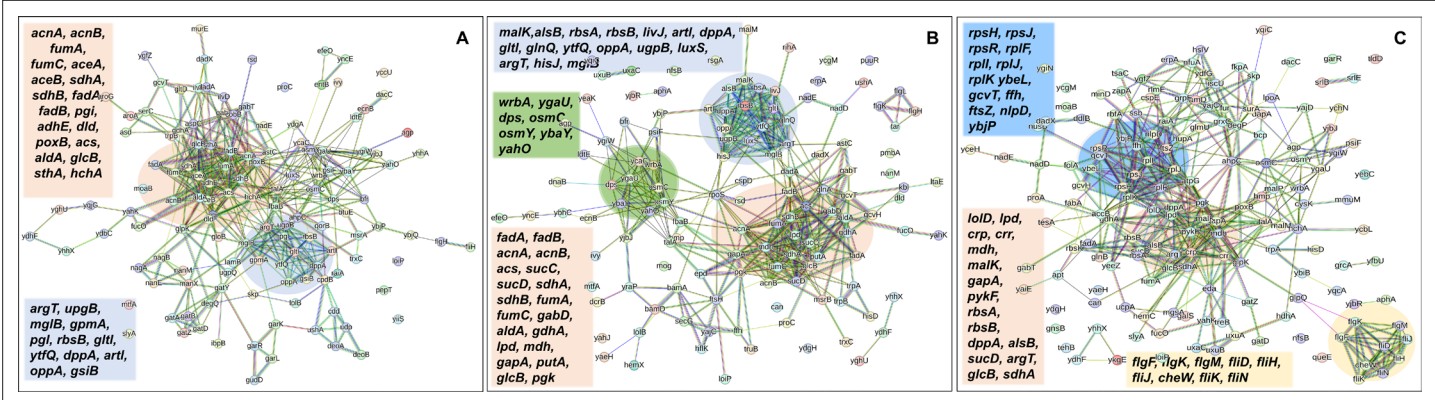

**Figure 6.** Proteomics data on the mutant strains, exhibiting increased gentamicin tolerance, indicate upregulation in proteins linked to energy metabolism. Cells from both wild-type and mutant strains at the mid-exponential phase (*t* = 3.5 hr) were collected after which protein extraction and digestion were carried out for mass spectrometry analysis. The STRING visual network depicts upregulated protein interactions of Δ*sucA* (**A**), Δ*gltA* (**B**), and Δ*nuoI* (**C**) mutants, compared to those of the wild-type strain. The protein clusters and their corresponding gene names are visually distinguished through color-coding on the networks. *N* = 3.

data transformation, normalization, and statistical procedures, as detailed elsewhere (*Aguilan et al., 2020*). Proteins exhibiting a positive log₂(fold change) and a p-value less than 0.05 were categorized as significantly upregulated while those displaying a negative log₂(fold change) and a p-value less than 0.05 were categorized as significantly downregulated in the mutant strains compared to the wild type (see *Supplementary file 2a–d* for upregulated proteins, and *Supplementary file 2e–h* for downregulated proteins). The STRING database was employed to predict both the physical and functional interactions between proteins by inputting the upregulated and downregulated proteins into the STRING interaction network (*Szklarczyk et al., 2023*). STRING carries out an automated pathway-enrichment analysis, focusing on the proteins entered and identifying pathways that occur more frequently than expected. The analysis is based on the statistical background of the entire genome and encompasses Gene Ontology annotations (all three domains), KEGG (Kyoto Encyclopedia of Genes and Genomes) pathways, Uniprot keywords, and the hierarchical clustering of the STRING network itself (see *Supplementary file 2i–p* for significantly altered pathways identified). When discussing our findings, we primarily reference the STRING network, as it offers the advantage of broader coverage, including potential novel modules that might not yet be classified as pathways (*Szklarczyk et al., 2019*).

In the context of the upregulated protein–protein association networks in the Δ*sucA* and Δ*gltA* mutant strains, the STRING analysis unveiled significant functional enrichments (*Supplementary file 2i ,j*) in the TCA cycle, carbon and pyruvate metabolism, formate c-acetyltransferase activity, and the fatty acid metabolic pathway (*Figure 6A, B* and *Supplementary file 2i, j*). Specifically, key proteins associated with these enrichments encompassed FumA, FumC, AcnA, AcnB, SdhA, SdhB, Dld, TalA, FbaB, FadA, FadB, and Acs (*Figure 6A, B*). Additionally, the Δ*sucA* and Δ*gltA* mutants exhibited an upregulated cluster of membrane proteins, particularly from the ABC transporter family (MalK, RbsA, RbsB, DppA, OppA, UgpB, and LuxS), with some proteins specialized in amino acid transport across the plasma membrane (ArtL, LivJ, GltI, GlnQ, ArgT, and HisJ) (*Figure 6A, B*). In the case of the Δ*gltA* mutant, a smaller network of upregulated stress-induced proteins was also observed, related to osmotic stress, oxidative stress, and starvation (OsmC, OsmY, YbaY, YgaU, YjbJ, Dps, and WrbA) (*Figure 6B*). Regarding the Δ*nuoI* mutant, the STRING analysis identified an upregulated functional network associated with carbon and pyruvate metabolism, featuring proteins SucD, FumA, SdhA, Crp, Eda, GalS, Crr, Mdh, PykF, Lpd, GapA, Pgk, PoxB, and TalA, as well as upregulated flagellar proteins (FlgF, FlgK, FlgM, FliD, FliH, FliJ, FliK, FliN, and CheW) (*Figure 6C* and *Supplementary file 2k*). Interestingly, our proteomics data on these three mutant strains, which display higher gentamicin tolerance, reveal upregulation in proteins associated with energy metabolism (e.g., TCA cycle, pyruvate metabolism) (*Figure 6*). This suggests that these mutant strains compensate for metabolic perturbations by enhancing the TCA cycle to preserve their ATP levels, redox activities, and PMF. Indeed, this proteomics data aligns well with measurements of ATP, RSG, and PMF highlighted in previous sections (*Figures 3–5*).

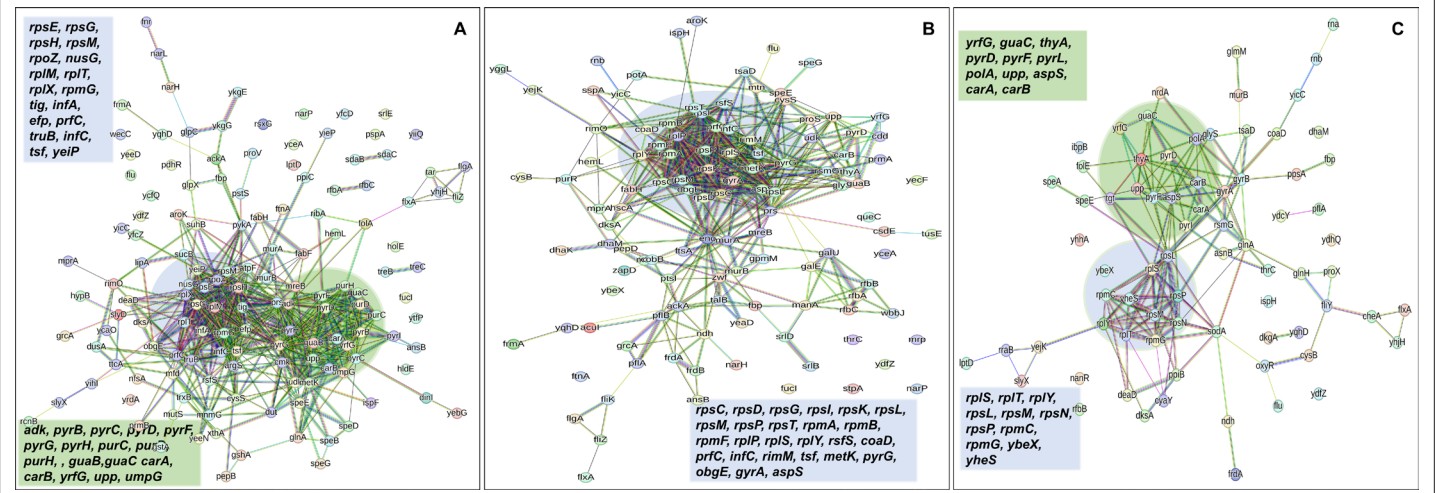

**Figure 7.** Proteomics data on the mutant strains, exhibiting increased gentamicin tolerance, reveal downregulation in proteins associated with ribosomes. The STRING visual network displays downregulated protein interactions for ΔsucA (**A**), ΔgltA (**B**), and Δnuol (**C**) mutants. *N* = 3.

The online version of this article includes the following figure supplement(s) for figure 7:

**Figure supplement 1.** The proteomic analysis of the gentamicin-sensitive Δicd mutant strain did not show a notable increase in energy metabolism or a decrease in proteins associated with ribosomes, in contrast to the observed patterns in ΔsucA, ΔgltA, and Δnuol mutants.

In the context of downregulated protein networks, the substantial functional enrichments unveiled through analysis using the STRING database were found to be intricately associated with both the large and small ribosomal subunits, denoted by the specific ribosomal proteins RplX, RplM, RplT, RplY, RplP, RplS, RpsE, RpsG, RpsH, RpsM, RpsT, Rpsl, RpsP, RpsK, RpsC, RpsD, RpsL, RpmG, RpmF, and RpmB (*Figure 7A–C* and *Supplementary file 2m–o*). These enrichments extended to domains encompassing translation factor activities, translation processes, and protein export mechanisms, featuring proteins RimM, InfA, InfC, Tsf, YeiP, NusG, Efp, PrfC, Mfd, GlyS, CysS, AspS, ArgS, and Tig (*Figure 7A–C*). Additionally, a noteworthy connection was established with the ribonucleoside monophosphate biosynthetic process, as characterized by proteins CarA, CarB, PyrD, PyrF, and PyrL (*Figure 7A–C*). The consistent downregulation of ribosomal binding proteins, ribosome biogenesis, translation factor activity, and the biosynthesis of ribonucleoside monophosphates is observed in the three mutant strains (ΔsucA, ΔgltA, and Δnuol), providing an explanation for the observed aminoglycoside tolerance in these mutants (*Figure 7A–C*). However, genome-level proteomics trends, specifically the drastic upregulation in energy metabolism and the drastic downregulation in ribosomal proteins, were not observed in the Δicd mutant strain (*Figure 7—figure supplement 1* and *Supplementary file 2l and p*). This may be expected, as the Δicd mutant strain did not consistently exhibit an increased tolerance trend to aminoglycosides compared to ΔsucA, ΔgltA, and Δnuol.

## Discussion

Our study aimed to explore the impact of gene knockouts associated with the TCA cycle and NADH dehydrogenase enzyme on aminoglycoside tolerance in *E. coli* MG1655. Various knockout strains showed increased tolerance to streptomycin, gentamicin, and amikacin compared to the wild-type strain. However, mutants like Δicd, ΔacnB, and ΔfumA did not consistently exhibit enhanced tolerance, highlighting the complex relationship between energy metabolism and antibiotic tolerance. The increased tolerance observed in certain TCA cycle gene deletions to gentamicin was also documented in a prior study (*Shan et al., 2015*). Our investigation further focused on four selected mutant strains (*sucA*, *gltA*, *nuol*, and *icd*), revealing time-dependent profiles of antibiotic-tolerant cells, with a substantial increase observed in the ΔsucA, ΔgltA, and Δnuol mutants during the mid-exponential growth phase. The observed tolerance was not linked to altered cell growth, and it appeared to be transient or reversible, as all mutants regained sensitivity to aminoglycosides during the lag phase of growth.

We further investigated the factors influencing aminoglycoside tolerance in *E. coli* MG1655 by examining the role of energy-dependent drug uptake, membrane potential, and genetic perturbations. Studies have shown that cells tolerant to antibiotics can transition to susceptibility to aminoglycosides through the metabolism of specific carbon sources (*Allison et al., 2011*). This shift in susceptibility is attributed to increased drug uptake, facilitated by increased activity in the ETC and membrane potential, both triggered by the breakdown of these particular carbon sources (*Allison et al., 2011*). Aminoglycoside uptake involves a three-step process: initial ionic binding to cells, followed by two energy-dependent phases (EDP I and EDP II) (*Taber et al., 1987*). The first step, concentration-dependent ionic binding, is not affected by inhibitors of energized uptake (*Taber et al., 1987*). The EDP I phase is gradual, requiring aminoglycoside concentration and a membrane potential for substantial uptake (*Taber et al., 1987*). Elevated aminoglycoside concentrations can bypass EDP I. EDP II involves rapid transport across the cytoplasmic membrane, relying on energy from electron transport and potentially ATP hydrolysis. Once inside the cell, studies suggest bactericidal effects through mistranslation and misfolding of membrane proteins, triggering oxidative stress and hydroxyl radical formation, ultimately leading to cell death (*Kohanski et al., 2008*; *Kohanski et al., 2007*). Despite hypothesizing that reduced PMF and aminoglycoside uptake might be linked to genetic perturbations, the mutant strains with increased tolerance showed no consistent pattern in metabolic activities in our study. Analysis of ATP levels, intracellular pH, and fluorophore-labeled aminoglycoside uptake did not reveal a clear association with antibiotic tolerance. Certainly, our proteomic analysis revealed significant enhancements in upregulated protein networks linked to the TCA cycle, carbon metabolism, pyruvate metabolism, and fatty acid pathways in the mutant strains exhibiting increased gentamicin tolerance. This sheds light on how these mutants manage and maintain their ATP levels, redox activities, PMF, and drug uptake, overcoming genetic disruptions in their metabolism. However, it is crucial to emphasize that our study does not refute the PMF-dependent mechanism of aminoglycoside uptake. The observation of similar drug uptake across all strains tested, including mutants and wild type, can be attributed to the ability of the mutant strains to maintain their PMF under the conditions studied here.

*Bruni and Kralj, 2020* postulated that the bactericidal action triggered by membrane potential might involve mechanisms beyond drug uptake, potentially influenced by the combined effects of pore formation by mistranslated proteins and membrane hyperpolarization. Additionally, their study proposed that the sudden shift in energy demand resulting from ribosome dissociation during aminoglycoside treatment could boost cellular ATP flux, leading to hyperpolarization through the concerted activity of NADH dehydrogenase and a reversed F1F0-ATPase (*Bruni and Kralj, 2020*). Interestingly, while our study documented dysregulation in membrane potential after gentamicin treatment, no significant differences were noted between gentamicin-sensitive and tolerant strains. Despite prior research implicating increased survival of specific ATP synthase knockouts in response to aminoglycoside treatments (*Bruni and Kralj, 2020*), our study did not report a clear trend regarding the gentamicin tolerance of these knockout strains. We acknowledge that variations in experimental conditions, such as medium composition, bacterial strains, growth phase, treatment time, duration, and antibiotic compositions, may account for the observed differences. Additionally, we recognize that genetic perturbations can have pleiotropic effects, potentially altering cell survival and death mechanisms, which may differ from those observed in the wild-type strain.

We conducted untargeted mass spectrometry to quantify proteins in the TCA cycle and ETC mutants. Utilizing the STRING database, we predicted functional protein interactions, revealing significant enrichments in downregulated protein networks associated with ribosomal subunits, translation factor activities, protein export mechanisms, and ribonucleoside monophosphate biosynthesis in the mutant strains that displayed higher gentamicin tolerance. The altered levels of ribosomal proteins observed in our mutant strains align well with a prior study that identified respiratory complex I in *E. coli* as a critical mutational target for promoting persister cell formation during the transition to a stationary phase (*Van den Bergh et al., 2022*). In that study, point mutations in respiratory complex I, responsible for proton translocation, were found to induce antibiotic tolerance (*Van den Bergh et al., 2022*). While they showed that mutations compromised proton translocation, key components such as NADH oxidation, electron transfer, and drug uptake remained largely unchanged (*Van den Bergh et al., 2022*). They also demonstrated that the increased persistence correlated with an acidified cytoplasm during the stationary phase, impacting protein translation and contributing to increased

antibiotic tolerance, yet no significant differences in pH and persistence were noted between mutant and wild-type strains during the exponential phase (*Van den Bergh et al., 2022*). Similarly, our investigation revealed that single gene deletions did not alter cellular PMF, redox activities, membrane potential, and drug uptake. However, the downregulation of ribosomal protein levels reported in our study may not be attributed to a reduction in cytoplasmic pH, as we did not observe alterations in pH levels in the knockout strains compared to the wild type during the mid-exponential phase. Certain genes linked to the deletion strains in our study, such as *icd*, *sucA*, *mdh*, and *fumA*, do not encode membrane-bound respiratory proteins. Strain-specific differences can significantly influence the outcomes and responses to various experimental conditions. Each strain of *E. coli* or any other microorganism can harbor unique genetic backgrounds, mutation profiles, and physiological characteristics, leading to distinct survival strategies in response to environmental stresses, such as antibiotic exposure. This variability is particularly evident in the *icd* deletion strain, as evidenced by its increased ciprofloxacin persistence in an independent study conducted by Lewis' group (*Manuse et al., 2021*). Furthermore, the same group demonstrated that the *icd* deletion strain exhibited increased sensitivity to gentamicin in another study (*Shan et al., 2015*), consistent with our observation here.

In summary, our study revealed that deletions in TCA cycle and ETC genes, including *sucA*, *gltA*, *mdh*, *sdhC*, *icd*, *acnB*, *fumA*, *nuoM*, and *nuoI*, increased aminoglycoside tolerance. Analyses, including flow cytometry and proteomics, demonstrated enhanced tolerance without significant changes in energy-dependent drug uptake or membrane potential. Our pathway analysis identified a downregulation in large and small ribosomal binding proteins, ribosome biogenesis, translation factor activity, and ribonucleoside monophosphate biosynthesis in all mutant strains, offering a plausible explanation for the observed aminoglycoside tolerance in these mutants. Altogether, our findings underscore the complexity of energy metabolism's role in antibiotic tolerance, providing valuable insights into the molecular mechanisms of aminoglycoside tolerance in specific mutant strains.

## Materials and methods

**Key resources table**

| Reagent type (species) or resource | Designation | Source or reference | Identifiers | Additional information |
|---|---|---|---|---|
| Strain, strain background (*Escherichia coli*) | *Escherichia coli* MG1655 Δ*sucA* mutant | other | | Ngo HG, *Ngo et al., 2024*. Unraveling Crp/cAMP-mediated metabolic regulation in *Escherichia coli* persister cells. *bioRxiv.* |
| Strain, strain background (*Escherichia coli*) | *Escherichia coli* MG1655 Δ*gltA* mutant | other | | *Ngo et al., 2024* |
| Strain, strain background (*Escherichia coli*) | *Escherichia coli* MG1655 Δ*nuoI* mutant | other | | *Ngo et al., 2024* |
| Strain, strain background (*Escherichia coli*) | *Escherichia coli* MG1655 Δ*icd* mutant | other | | *Ngo et al., 2024* |
| Strain, strain background (*Escherichia coli*) | *Escherichia coli* MG1655 Δ*sdhC* mutant | other | | *Ngo et al., 2024* |
| Strain, strain background (*Escherichia coli*) | *Escherichia coli* MG1655 Δ*mdh* mutant | other | | *Ngo et al., 2024* |
| Strain, strain background (*Escherichia coli*) | *Escherichia coli* MG1655 Δ*acnB* mutant | other | | *Ngo et al., 2024* |

*Continued on next page*

*Continued*

| Reagent type (species) or resource | Designation | Source or reference | Identifiers | Additional information |
|---|---|---|---|---|
| Strain, strain background (*Escherichia coli*) | *Escherichia coli* MG1655 Δ*fumA* mutant | other | | ***Ngo et al., 2024*** |
| Strain, strain background (*Escherichia coli*) | *Escherichia coli* MG1655 Δ*nuoM* mutant | other | | ***Ngo et al., 2024*** |
| Strain, strain background (*Escherichia coli*) | *Escherichia coli* MG1655 Δ*atpA* mutant | other | | ***Ngo et al., 2024*** |
| Strain, strain background (*Escherichia coli*) | *Escherichi. coli* MG1655 Δ*atpB* mutant | other | | ***Ngo et al., 2024*** |
| Strain, strain background (*Escherichia coli*) | *Escherichia coli* MG1655 Δ*atpC* mutant | other | | ***Ngo et al., 2024*** |
| Strain, strain background (*Escherichia coli*) | *Escherichia coli* MG1655 Δ*atpD* mutant | other | | ***Ngo et al., 2024*** |
| Strain, strain background (*Escherichia coli*) | *Escherichia coli* K-12 BW25113 Δ*atpA* mutant | Horizon Discovery | Catalog # OEC4988 | |
| Strain, strain background (*Escherichia coli*) | *Escherichia coli* K-12 BW25113 Δ*atpB* mutant | Horizon Discovery | Catalog # OEC4988 | |
| Strain, strain background (*Escherichia coli*) | *Escherichia coli* K-12 BW25113 Δ*atpC* mutant | Horizon Discovery | Catalog # OEC4988 | |
| Strain, strain background (*Escherichia coli*) | *Escherichia coli* K-12 BW25113 Δ*atpD* mutant | Horizon Discovery | Catalog # OEC4988 | |
| Strain, strain background (*Escherichia coli*) | *Escherichia coli* K-12 BW25113 Δ*atpE* mutant | Horizon Discovery | Catalog # OEC4988 | |
| Strain, strain background (*Escherichia coli*) | *Escherichia coli* K-12 BW25113 Δ*atpF* mutant | Horizon Discovery | Catalog # OEC4988 | |
| Strain, strain background (*Escherichia coli*) | *Escherichia coli* K-12 BW25113 Δ*atpG* mutant | Horizon Discovery | Catalog # OEC4988 | |
| Strain, strain background (*Escherichia coli*) | *Escherichia coli* K-12 BW25113 Δ*atpH* mutant | Horizon Discovery | Catalog # OEC4988 | |
| Strain, strain background (*Escherichia coli*) | *Escherichia coli* K-12 BW25113 Δ*atpI* mutant | Horizon Discovery | Catalog # OEC4988 | |
| Recombinant DNA Reagent | pGFPR01 | ***Martinez et al., 2012*** | | |
| Commercial assay or kit | BacTiter-Glo Microbial Cell Viability assay | Promega Corporation | Catalog # G8230 | |
| Commercial assay or kit | *Bac*Light Redox Sensor Green Vitality kit | Thermo Fisher | Catalog # B34954 | |
| Chemical compound or drug | Gentamicin-Texas Red Conjugate | AAT Bioquest, Inc | Catalog # 24300 | |

*Continued*

| Reagent type (species) or resource | Designation | Source or reference | Identifiers | Additional information |
|---|---|---|---|---|
| Chemical compound or drug | Polymyxin B Sulfate | Millipore Sigma | Catalog # 52-91-1GM | |
| Chemical compound or drug | DiSC3(5) (3,3'-Dipropylthiadicarbocyanine Iodide) | TCI America | Catalog # D4456 | |
| Commercial assay or kit | MIC Test Strips | Fisher Scientific | | |
| Software, algorithm | Prism (version 10.1.2) | GraphPad | RRID: SCR_002798 | http://www.graphpad.com/ |
| Software, algorithm | FlowJo (version 10.10.0) | Becton, Dickinson & Company | RRID: SCR_008520 | https://www.flowjo.com/solutions/flowjo |

## Bacterial strains, chemicals, media, and growth conditions

We employed the in-house *E. coli* K-12 MG1655 strain for our experiments. The gene deletions in *E. coli* K-12 MG1655 were established in our prior studies (**Ngo et al., 2024**) utilizing the Datsenko and Wanner method (**Datsenko and Wanner, 2000**). *E. coli* K-12 BW25113 and its corresponding knockout strains were procured from the Keio collection, purchased from Horizon Discovery (Catalog # OEC4988, Lafayette, CO). The validity of all deletions was confirmed through the use of check primers (**Ngo et al., 2024**). All chemicals utilized in the study were purchased from Fisher Scientific (Atlanta, GA) or VWR International (Pittsburg, PA) unless stated otherwise. The Gentamicin-Texas Red Conjugate (Catalog # 24300) was purchased from AAT Bioquest, Inc (CA, USA). The ATP measurement kit (Catalog # G8230) was acquired from Promega Corporation (Madison, WI). Standard LB broth was prepared by dissolving 5 g yeast extract, 10 g tryptone, and 10 g sodium chloride in 1 l of deionized (DI) water. LB agar was made by dissolving 40 g of pre-mixed LB agar powder in 1 l DI water. Sterilization of LB broth and LB agar was done via autoclaving at 121°C and 103.421 kPa. To determine tolerant cells in cultures, gentamicin (50 µg/ml), streptomycin (50 µg/ml), and amikacin (50 µg/ml) were used, and their concentrations were selected to be much higher than the MICs. Whenever specified, gentamicin at various concentrations (e.g., 5 µg/ml) was also tested. MICs of antibiotics for *E. coli* (**Andrews, 2001**) were determined using MIC Test strips (Fisher Scientific) and the results are presented in *Supplementary file 1a*. Ampicillin (100 µg/ml) was utilized to maintain plasmids in the cells. 0.2% L-arabinose at 13.3 mM was added to the media to induce GFP expression. To decrease the antibiotic concentration below the MIC, a sterile 1× phosphate-buffered saline (PBS) solution was utilized to wash the cells. Antibiotics were dissolved in DI water and sterilized using a 0.2-µm PES (polyether sulfone) syringe filter. Overnight cultures were prepared in 14 ml round-bottom tubes (Catalog # 14-959-1B, Fisher Scientific) by incubating cells from a frozen 25% glycerol stock (−80°C) in 2 ml of LB at 37°C and 250 rpm. After 24 hr, cells from the overnight cultures were diluted 1:100-fold in 2 ml of fresh LB medium in 14 ml round bottom test tubes or 25 ml LB in 250 ml baffled flasks and cultured further to achieve the desired growth phase for the assays.

## Cell growth assay

To prepare overnight precultures, cells from frozen stocks were inoculated into 2 ml of LB medium in 14 ml round-bottom tubes and cultured for 24 hr at 37°C with shaking at 250 rpm. The overnight precultures were then diluted 100-fold in 14 ml round-bottom tubes containing 2 ml of LB medium and incubated in the shaker at 250 rpm and 37°C. The growth of the cultures was determined by measuring the number of cells per ml using flow cytometry. To perform this task, cells were transferred to PBS at specified time points and analyzed with a flow cytometer (NovoCyte Flow Cytometer, NovoCyte 3000RYB, ACEA Biosciences Inc, San Diego, CA). The cell populations were delineated on flow diagrams utilizing the forward and side scatter parameters, with controls including PBS with cells

and PBS without cells. The inclusion of a PBS solution without cells aids in noise determination. The instrument is capable of quantifying both the number of events and the volume of the solution under analysis. Roughly 30,000–50,000 events were analyzed for each sample.

## Clonogenic survival assay

Overnight cultures were prepared in 14 ml round-bottom test tubes by incubating cells from a frozen 25% glycerol stock (−80°C) in 2 ml of LB at 37°C and 250 rpm. After 24 hr, cells from the overnight cultures were diluted 1:100-fold in 2 ml of fresh medium and cultured in the incubator at 37°C with shaking. At the designated growth phase or time points, cells were exposed to antibiotics at the specified concentrations (5 or 50 µg/ml gentamicin, 50 µg/ml streptomycin, and 50 µg/ml amikacin) for a duration of 5 hr. To determine the number of live cells before antibiotic exposure, 10 µl of cell cultures were serially diluted in PBS and plated on an LB agar plate, which was incubated for 16 hr at 37°C. During the antibiotic treatments, 1 ml cultures were collected after 5 hr and washed twice with PBS through centrifugation at 13,300 rpm (17,000 × $g$) for 3 min to remove antibiotics. After the final centrifugation, 900 µl of supernatant was removed using a pipette, and the cell pellets were resuspended in the remaining 100 µl of PBS. Next, 10 µl of the cell suspensions were serially diluted in PBS, and 10 µl of the diluted cell suspensions were spotted onto LB agar plates. The plates were then incubated at 37°C for at least 16 hr, and the colony-forming units (CFU) were counted to determine the number of live cells present in the cultures. Survival fractions were calculated by dividing the number of surviving cells (after treatment) by the initial number of cells (before treatment).

To generate survival time profiles, cells from the overnight cultures were diluted 1:100 in 25 ml of fresh LB medium in 250 ml baffled flasks and incubated in the incubator at 37°C with shaking. At the designated time points, 2 ml of cell cultures were transferred to 14 ml round-bottom test tubes and subjected to antibiotic treatment at the specified concentration (50 µg/ml gentamicin) for a duration of 5 hr. Following treatment, cells were collected, washed to remove antibiotics, and plated for CFU enumeration, as previously described.

## RSG dye staining

To measure bacterial reductase and ETC activities, we used the *Bac*Light Redox Sensor Green Vitality kit (Catalog # B34954, Thermo Fisher) following the manufacturer's instructions. Overnight cultures were prepared in 14 ml Falcon tubes by incubating cells from a frozen 25% glycerol stock (−80°C) in 2 ml of LB at 37°C and 250 rpm. After 24 hr, cells from the overnight cultures were diluted 1:100-fold in 2 ml of fresh medium and cultured in the incubator at 37°C with shaking. For analyzing the cell populations during mid-exponential and early stationary phases ($t$ = 3, 4, 5, and 6 hr), we diluted the cells in 1 ml of 0.85% sodium chloride solution in flow cytometry tubes (5 ml round bottom Falcon tubes) by varying amounts (10-, 20-, 20-, and 50-fold, respectively). After that, RSG dye was added to the cells at a concentration of 1 µM, and the samples were incubated at 37°C for 10 min before flow cytometry analysis. For the negative controls, cell suspensions were treated with 20 µM CCCP 5 min prior to RSG staining to disrupt membrane electron transport (*Figure 3—figure supplement 1*). Positive controls consisted of mid-exponential phase cells. The cell populations were gated on flow diagrams using the forward and side scatter parameters of unstained controls. Cells were excited at 488 nm with a solid-state laser, and green fluorescence was collected with a 530/30 bandpass filter.

## ATP measurement

The BacTiter-Glo Microbial Cell Viability assay kit (Catalog # G8230, Promega Corporation) was used to measure the intracellular ATP levels of both *E. coli* MG1655 WT and mutant strains during a specified growth phase, following the manufacturer's instructions. To generate a standard curve, ATP solutions of known concentrations were used (*Figure 2*, *Figure 3—figure supplement 2*). Background luminescence was measured using LB broth.

## pH measurement

For pH measurements in *E. coli*, a pGFPR01 plasmid was used, in which the GFP derivative ratiometric pHluorin is expressed from the arabinose-induced promoter $P_{BAD}$. This plasmid was kindly provided by Keith A. Martinez II (Department of Biology, Kenyon College, Gambier, Ohio). A comprehensive pH measurement protocol was obtained from a prior study (*Van den Bergh et al., 2022*). Overnight

cultures of WT and mutant strains were prepared in 14 ml tubes by incubating cells carrying the pGFPR01 plasmid from a frozen 25% glycerol stock (−80°C) in 2 ml of LB, 0.2% L-arabinose and 100 mg/ml ampicillin, at 37°C and 250 rpm. After 24 hr, cells from the overnight cultures were diluted 1:100-fold in 2 ml of fresh medium, 0.2% L-arabinose and 100 mg/ml ampicillin and cultured in the incubator at 37°C with shaking. Fluorescence was measured at 410 and 470 nm using a plate reader at different time points, after measuring and normalizing the optical density of cells in liquid culture. The 410/470 fluorescence ratios were recorded to determine the cytoplasmic pH using the standard curve (*Figure 3—figure supplement 3*). To generate standard pH versus fluorescence ratio curves for *E. coli* MG1655 cells, transmembrane pH was collapsed by adding 40 mM potassium benzoate and 40 mM methylamine hydrochloride to the cells, equalizing the difference between the external and internal pH. The cultures were then buffered to different pH levels ranging from 5 to 10 using a 50 mM concentration of 2-(*N*-morpholino) ethanesulfonic acid or 3-(*N*-morpholino) propanesulfonic acid, and corresponding fluorescence values at 410 and 470 nm were obtained using a plate reader. The Boltzmann equation was used to establish the standard curve for each bacterial strain based on the provided data (*Martinez et al., 2012*).

## DiSC$_3$(5) assay

The fundamental mechanism of this assay and the assessment of PMF components using DiSC$_3$(5) have been extensively detailed elsewhere (*Farha et al., 2013*; *Panta and Doerrler, 2020*; *Stokes et al., 2020*). *E. coli* MG1655 wild-type and mutant cells in both exponential and early stationary phases (at time points $t = 3, 4, 5$, and 6 hr) were collected and subjected to two washes with an assay buffer containing 5 mM HEPES (N-[2-Hydroxyethyl]piperazine-N-[2-ethanesulfonic] hemisodium salt) and 20 mM glucose. The cell density was set to OD$_{600}$=0.1, and the cells were stained with 1 µM DiSC$_3$(5). Fluorescence readings were taken at specified intervals using a plate reader, with excitation and emission wavelengths set at 620 and 670 nm, respectively. Gentamicin (5 µg/ml, 50 µg/ml, or other specified concentrations) was introduced 20 min after cells reached equilibrium. At this point, the probe was released into the medium, leading to an upsurge in fluorescence. Polymyxin B (32 µg/ml) was included as controls and administered 20 min after equilibrium to dissipate the electron gradient of PMF. The concentration of polymyxin B was determined based on a prior study (*Stokes et al., 2020*).

## Gentamicin uptake assay

Overnight cultures were prepared in 14 ml tubes by incubating cells from a frozen 25% glycerol stock (−80 °C) in 2 ml of LB at 37°C and 250 rpm. After 24 hr, cells from the overnight cultures were diluted 1:100-fold in 2 ml of fresh medium and cultured in the incubator at 37°C with shaking. At the mid-exponential phase ($t = 3.5$ hr), 100 µl of cell cultures for both wild type and mutants were exposed to GTTR at a final concentration of 25 or 5 µg/ml. Untreated mid-exponential phase cells were used as negative controls. The samples containing GTTR were then incubated for 1 hr at 37°C with shaking at 250 rpm. Subsequently, 20 µl from each sample was washed with 500 µl of PBS (1×). After the final washing step, cell pellets were resuspended in 500 µl of PBS. All samples were subjected to analysis using a flow cytometer equipped with lasers emitting light at a wavelength of 561 nm, and the resulting red fluorescence was detected using a 615/20 nm bandpass filter.

## Proteomics sample preparation and analysis

Cells from both wild-type and mutant strains at the mid-exponential phase ($t = 3.5$ hr) were collected after which protein extraction and digestion were carried out for liquid chromatography–mass spectrometry (LC–MS) analysis by following the Sample Preparation by Easy Extraction and Digestion (SPEED) procedure. The experiments were conducted at the University of Houston Mass Spectrometry Laboratory under a service fee. Comprehensive details regarding the protein isolation and digestion methods can be found elsewhere (*Doellinger et al., 2020*). Briefly, cell pellets were added 20 µg trifluoroacetic acid and incubated at room temperature for 5 min followed by addition of 200 µl of 2 M trisbase. After adding Tris(2-carboxyethyl) phosphine (10 mM) and 2-chloroacetamide (40 mM), the reaction mixture was heated at 95°C for 5 min. Digestion was performed by adding trypsin (1/40, wt/wt) and incubation at 37°C overnight. The digested peptides were cleaned up using a C18 Ziptip and vacuum dried using a CentriVap (Labconco, Kansas City, MO). Each dried sample was resuspended

in 2% acetonitrile (ACN) with 0.1% formic acid (FA) for LC–MS analysis. The method involving LC–MS has been detailed in a separate publication (*Qin et al., 2022*). Specifically, a NanoElute LC system connected to a timsTOF Pro (Bruker Daltonics, Germany) through a CaptiveSpray source was utilized. Samples were loaded onto an in-house packed column (75 μm × 20 cm, 1.9 μm ReproSil-Pur C18 particle from Dr. Maisch GmbH, Germany) with a column temperature of 40°C. Mobile phases included buffer A (0.1% FA in water) and buffer B (0.1% FA in ACN). The short gradient was 0–17.8 min, from 2% B to 30% B, followed by 18.3 min to 95% B, and 20.7 min to 95% B. The parallel accumulation-serial fragmentation (PASEF) mode with 4 PASEF scans per cycle was employed. The electrospray voltage was set at 1.4 kV, and the ion transfer tube temperature was maintained at 180°C. Full MS scans were conducted across the mass-to-charge (*m/z*) range of 150–1700. The target intensity value was $2.0 \times 10^5$ with a threshold of 2,500. A fixed cycle time of 0.53 s was established, and a dynamic exclusion duration of 0.4 min with a ±0.015 amu tolerance was applied. Only peaks with a charge state of ≥2 were chosen for fragmentation. The default settings of MSFragger, a database search tool designed for peptide identification in MS-based proteomics (*Yu et al., 2020*), were applied to analyze data obtained from the mentioned instrument. The UniProt-SwissProt *E. coli* K12 database (Taxon ID 83333, downloaded on 6/19/2023, 4518 entries) served as the reference. Fixed modification involved cysteine carbamidomethylation, while variable modifications included methionine oxidation and acetylation. Peptide length was restricted to 7–50, allowing for 2 missed cleavages. Both precursor and product ion masses were set as monoisotopic. The false discovery rate was controlled at <1% at the peptide spectrum match, peptide, and protein levels. The raw data of proteomics analysis are provided in *Supplementary file 3*. The mass spectrometry proteomics data have been deposited to the ProteomeXchange Consortium via the PRIDE (*Perez-Riverol et al., 2022*) partner repository with the dataset identifier PXD053082.

## Proteomics data analysis

The processing of proteomics data and the calculations of fold change were essentially carried out following the methods described in the paper by *Aguilan et al., 2020*. In summary, we utilized Excel spreadsheets for key stages of data transformation, normalization, fold change, and p-value calculation. Initially, proteins lacking quantitative values were excluded, and a logarithm transformation was applied to achieve a normal distribution of data. Normalization, using both average and slope methods, was then employed to minimize intragroup variation in technical replicates and log fold change calculations compared to the transformed unnormalized data. Subsequently, missing values were imputed by replacing them with approximated values using the Probabilistic Minimum Imputation method. Following imputation, we determined the relative ratio of each protein in mutant and wild-type strains, along with p-value calculation using the parametric *t*-test. The selection of the *t*-test type involved an *F*-test to evaluate whether the replicates for each protein exhibited homoscedastic (equal variances) or heteroscedastic (unequal variances) characteristics. For the identification of significant networks among input proteins, we utilized the STRING tool V 12.0. This entailed inputting proteins that were significantly upregulated and downregulated based on specified thresholds, as detailed elsewhere (*Szklarczyk et al., 2023*).

## Statistical analysis

All assays were conducted using at least three independent biological replicates. The figures display the mean value and standard error for each data point. Statistical analysis was performed using GraphPad Prism software, with one-way analysis of variance with Dunnett's post hoc test to determine significance. The p-value threshold was set at $*p < 0.05$, $**p < 0.01$, $***p < 0.001$, and $****p < 0.0001$.

## Acknowledgements

The authors would like to thank the members of Orman Lab for their help. This study was supported by NSF CAREER 2044375 and NIH/NIAID R01-AI143643.

Proteomics experiments were conducted at the Mass Spectrometry Laboratory of Dr. Chengzhi Cai at the University of Houston, with the associated service fees.

## Additional information

### Funding

| Funder | Grant reference number | Author |
|---|---|---|
| National Science Foundation | 2044375 | Mehmet A Orman |
| National Institute of Allergy and Infectious Diseases | AI143643 | Mehmet A Orman |
| NSF CAREER | 2044375 | Mehmet A Orman |
| NIH/NIAID | R01-AI143643 | Mehmet A Orman |

The funders had no role in study design, data collection, and interpretation, or the decision to submit the work for publication.

### Author contributions

Rauf Shiraliyev, Conceptualization, Formal analysis, Validation, Investigation, Methodology, Writing – original draft, Writing – review and editing; Mehmet A Orman, Conceptualization, Data curation, Supervision, Funding acquisition, Investigation, Writing – original draft, Project administration, Writing – review and editing

### Author ORCIDs

Rauf Shiraliyev (ID) http://orcid.org/0009-0008-9488-1633
Mehmet A Orman (ID) https://orcid.org/0000-0001-8499-9154

Reviewer #2 (Public Review): https://doi.org/10.7554/eLife.94903.3.sa1
Author response https://doi.org/10.7554/eLife.94903.3.sa2

## Additional files

### Supplementary files

• Supplementary file 1. The minimum inhibitory concentrations of aminoglycosides were assessed in both wild type and mutant *E. coli* MG1655 strains. (**a**) The minimum inhibitory concentration (MIC) levels of streptomycin, gentamicin, and amikacin were examined in tricarboxylic acid cycle (TCA) and electron transport chain (ETC) mutants as well as the wild type.

• Supplementary file 2. Proteomics data analysis identified upregulated and downregulated proteins in mutant *E. coli* MG1655 strains with subsequent pathway analysis for these proteins compared to the wild type. (**a**) Upregulated proteins at mid-exponential phase ($t$ = 3.5 hr) in the Δ*sucA* mutant strain relative to the wild type. FC: fold change. A significance threshold of $p < 0.05$, based on *F*- and *t*-statistics (see Materials and methods), is applied. (**b**) Upregulated proteins at mid-exponential phase ($t$ = 3.5 hr) in the Δ*gltA* mutant strain relative to the wild type. FC: fold change. A significance threshold of $p < 0.05$ is applied. (**c**) Upregulated proteins at mid-exponential phase ($t$ = 3.5 hr) in the Δ*nuoI* mutant strain relative to the wild type. FC: fold change. A significance threshold of $p < 0.05$ is applied. (**d**) Upregulated proteins at mid-exponential phase ($t$ = 3.5 hr) in the Δ*icd* mutant strain relative to the wild type. FC: fold change. A significance threshold of $p < 0.05$ is applied. (**e**) Downregulated proteins at mid-exponential phase ($t$ = 3.5 hr) in the Δ*sucA* mutant strain relative to the wild type. FC: fold change. A significance threshold of $p < 0.05$ is applied. (**f**) Downregulated proteins at mid-exponential phase ($t$ = 3.5 hr) in the Δ*gltA* mutant strain relative to the wild type. FC: fold change. A significance threshold of $p < 0.05$ is applied. (**g**) Downregulated proteins at mid-exponential phase ($t$ = 3.5 hr) in the Δ*nuoI* mutant strain relative to the wild type. FC: fold change. A significance threshold of $p < 0.05$ is applied. (**h**) Downregulated proteins at mid-exponential phase ($t$ = 3.5 hr) in the Δ*icd* mutant strain relative to the wild type. FC: fold change. A significance threshold of $p < 0.05$ is applied. (**i**) The pathway analysis for the upregulated proteins in the Δ*sucA* strain compared to the wild type. This analysis integrates statistical analysis across the entire genome and includes various functional pathway classification frameworks such as Gene Ontology annotations, KEGG pathways, Uniprot, and STRING. *Count in network:* The first number indicates how many proteins in our network are annotated with a particular term. The second number indicates how

many proteins in total (in our network and the background) have this term assigned. *Strength:* Log10(observed/expected). This measure describes how large the enrichment effect is. It is the ratio between (1) the number of proteins in our network that are annotated with a term and (2) the number of proteins that we expect to be annotated with this term in a random network of the same size. *False discovery rate:* This measure describes how significant the enrichment is. Shown are p-values corrected for multiple testing within each category using the Benjamini–Hochberg procedure. Note: When discussing our findings in the manuscript, we primarily reference the local network cluster (STRING), as it offers the advantage of broader coverage, including potential novel modules that might not yet be classified as pathways. (**j**) The pathway analysis for the upregulated proteins in the Δ*gltA* strain compared to the wild type. See the legend of *Supplementary file 2i* for further details. (**k**) The pathway analysis for the upregulated proteins in the Δ*nuoI* strain compared to the wild type. See the legend of *Supplementary file 2i* for further details. (**l**) The pathway analysis for the upregulated proteins in the Δ*icd* strain compared to the wild type. See the legend of *Supplementary file 2i* for further details. (**m**) The pathway analysis for the downregulated proteins in the Δ*sucA* strain compared to the wild type. See the legend of *Supplementary file 2i* for further details. (**n**) The pathway analysis for the downregulated proteins in the Δ*gltA* strain compared to the wild type. See the legend of *Supplementary file 2i* for further details. (**o**) The pathway analysis for the downregulated proteins in the Δ*nuoI* strain compared to the wild type. See the legend of *Supplementary file 2i* for further details. (**p**) The pathway analysis for the downregulated proteins in the Δ*icd* strain compared to the wild type. See the legend of *Supplementary file 2i* for further details.

• Supplementary file 3. The raw proteomics data included quantified protein levels in samples from both wild type and mutant strains.

• MDAR checklist

## Data availability

The proteomics data are provided in Supplementary file 3 and are submitted to one of the NIH829 designated repositories, PRIDE (*Perez-Riverol et al., 2022*), with identifier PXD053082. All data generated or analyzed during this study have been incorporated into Figures 1–7 and Supplementary Files 1–3.

The following dataset was generated:

| Author(s) | Year | Dataset title | Dataset URL | Database and Identifier |
|---|---|---|---|---|
| Shiraliyev A, Orman MA | 2024 | Metabolic disruption impairs ribosomal protein levels, resulting in enhanced aminoglycoside tolerance | https://www.ebi.ac.uk/pride/archive/projects/PXD053082 | PRIDE, PXD053082 |

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
