## [Editor Report · eLife assessment]

The findings of this study are **valuable** as they challenge the dogma regarding the link between lowered bacterial metabolism and tolerance to aminoglycosides. The authors propose that the well-known tolerance to AG of mutants such as those of complexes I and II is not due to a decrease in the proton motive force and thus antibiotic uptake. The results presented here are **convincing**.

---

## [Referee Report · Reviewer #2 (Public Review)]

Summary:

This interesting study challenges the dogma regarding the link between bacterial metabolism decrease and tolerance to aminoglycosides (AG). The authors demonstrate that mutants well-known for being tolerant to AG, such as those of complexes I and II, are not so due to a decrease in the proton motive force (PMF) and thus antibiotic uptake, as previously reported in the literature.

Strengths:

This is a complete study that employs several read-outs.

In this revised version, the authors have carefully addressed all the reviewers' comments. I appreciate the effort made in this new version to clarify that this study does not refute the PMF-dependent mechanism of aminoglycoside uptake (in the discussion_ lines 731-734_).

The addition of the requested experiments using lower concentrations of aminoglycosides is a considerable improvement as it allows for comparison with previously published results.

---

## [Author Response]

The following is the authors’ response to the original reviews.

**Reviewer #1 (Public Review):**
Summary:In this study, the authors investigate the tolerance of aminoglycosides in *Escherichia coli* mutants deleted in the Krebs cycle and respiratory chain enzymes. The motivation for this study is unclear. Transport of aminoglycosides is pmf-dependent, as the authors correctly note, and knocking out energy-producing components leads to tolerance of aminoglycosides, this has been well established. In S. aureus, clinically relevant "small colony" strains selected for in the course of therapy with aminoglycosides acquire null mutations in the biosynthesis of heme or ubiquinone, and have been studied in detail. In *E. coli*, such knockouts have not been reported in clinical isolates, probably due to severe fitness costs.

Response: We sincerely appreciate the time and consideration the reviewer dedicated to evaluating our manuscript. It's important to highlight that while the transport of aminoglycosides is PMF-dependent, recent studies underscore the potential role of metabolic mutations in antibiotic tolerance, a facet that warrants further investigation. For instance, the study by Henimann’s and Michiels' groups explored genomic changes in *E. coli* strains (including uropathogenic UTI89 strains) subjected to daily antibiotic exposure (Van den Bergh et al., 2022). Notably, mutations predominantly occurred in genes of the *nuo* operon, a key component of *E. coli* energy metabolism, suggesting a link between metabolic adaptations and antibiotic tolerance. Furthermore, the research by Collin's group revealed previously unrecognized genes related to central metabolism (e.g., *icd, gltD, sucA*) that contribute to antibiotic resistance in *E. coli* cells exposed to multiple antibiotics, including aminoglycosides (Lopatkin et al., 2021). These findings are corroborated by the presence of similar mutations in clinical *E. coli* pathogens, as evidenced by the analysis of a large library of 7243 *E. coli* genomes from NCBI Pathogen Detection (Lopatkin et al., 2021). The clinical relevance of metabolic mutations in antibiotic tolerance is increasingly recognized, yet their underlying mechanisms remain enigmatic. Therefore, elucidating the role of metabolic pathways in conferring antibiotic tolerance is highly critical. We have updated the introduction to clearly convey our motivation in this study (see page 3).

At the same time, single-cell analysis has shown that individual cells with a decrease in the expression of Krebs cycle enzymes are tolerant of antibiotics and have lower ATP (Manuse et al., PLoS Biol 19: e3001194). The authors of the study under review report that knocking out ICD, isocitrate dehydrogenase that catalyzes the rate-limiting step in the Krebs cycle, has little effect on aminoglycoside tolerance and actually leads to an increase in the level of ATP over time. This observation does not seem to make much sense and contradicts previous reports, specifically that *E. coli* ICD is tolerant of antibiotics and, not surprisingly, produces Less ATP (Kabir and Shimizu, Appl Micro-biol Biotechnol. 2004; 65(1):84-96; Manuse et al., PLoS Biol 19: e3001194). Mutations in other Krebs cycle enzymes, unlike ICD, do lead to a dramatic increase in tolerance of aminoglycosides according to the paper under review. This is all very confusing.

Response: Although our data cannot be directly compared to that of Kabir and Shimizu (Mohiuddin Kabir and Shimizu, 2004), due to the utilization of entirely different experimental procedures and measurement techniques, we can draw some parallels to the study conducted by Lewis’ group (Manuse et al., 2021), despite certain differences in experimental protocols. Furthermore, the reviewer has made strong assertions regarding our manuscript based on the findings of Lewis’ group. Thus, we believe it's pertinent to expand our response regarding that study.

In the study of Lewis’ group, bacterial cells were inoculated at a ratio of 1:100 into LB medium from an overnight culture (approximately 16 hours). Subsequently, the cultures were incubated at 37°C for approximately 2 hours, and ATP levels were measured using the BacTiter Glo kit (Promega, Madison, WI, USA). ATP levels were then normalized to cell density, determined through optical density measurements, and represented on a linear diagram. As demonstrated in Supplementary Figure S1c of their paper, there was a 10-15% reduction in normalized ATP levels in the *icd* mutant compared to the wild type. In our experiments, cells were grown for 24 hours in overnight cultures, diluted 100-fold in fresh media, and ATP levels were measured at 3, 4, 5, and 6 hours using the same kit. ATP levels were normalized to cell counts quantified by flow cytometry. Upon analyzing our data of the *icd* mutant for around 3 hours (the time point closest to that of the study of Lewis’ group), we observed a reduction of approximately 15-20% (without statistical significance) in the *icd* mutant compared to the wild-type (see raw data, linear plot, and logarithmic plot below; Author response image 1), which aligns with the findings of Lewis’ group.

We further investigated the gentamicin tolerance of both wild-type and *icd* mutant strains of *E. coli* BW25113 (Author response image 2). Our findings indicate that the increased sensitivity of the *icd* mutant of the MG1655 strain to gentamicin is similar to the observation in the other *E. coli* strain.

**Author response image 1. sa2fig1:** ATP levels in the *icd* mutant. ATP levels of both the mutant and wild-type strains were measured at t=3 hours of cell growth and normalized to cell counts. The figure presents the raw data (a), linear plot (b), and logarithmic plot (c) of the same dataset. This data corresponds to the first panel of Figure 3B in the manuscript.

**Author response image 2. sa2fig2:** Gentamicin tolerance of wild-type and *icd* mutant strains of *E. coli* BW25113. Both wild type and mutant strains were treated with gentamicin (50 µg/ml) for 5 hours at the mid-exponential phase. Cells were plated before and after treatment for CFU/ml counts. The dashed line represents the limit of detection. CFU: Colony forming units.

We think that there are two primary reasons why our study cannot contradict the findings of the Lewis group:

Firstly, our study cannot be directly compared to theirs, as they did not comprehensively explore the impact of gene deletions on cell metabolism beyond the measurement of ATP levels at a single time point (Manuse et al., 2021). Our study encompasses various metabolic parameters such as cellular ATP, redox status, proton motive force (PMF), intracellular pH, and drug uptake throughout the exponential and/or early stationary phase. Additionally, we conducted proteomic analysis for five different strains including mutants and wild type. Moreover, we performed pathway enrichment analysis grounded in the statistical background of the entire genome, encompassing various functional pathway classification frameworks such as Gene Ontology annotations, KEGG pathways, and Uniprot keywords. The results of these pathway enrichment analyses are now available in the Supplementary file 2 (see Supplementary file 2i-p). Thus, we believe it is unjust to deem our study contradictory compared to the Lewis group's study, which does not have a comprehensive analysis of the metabolism of the mutant strains they investigated.

Secondly, our study cannot be compared to that specific study (Manuse et al., 2021) due to the utilization of a distinct antibiotic (ciprofloxacin). Cell tolerance is heavily reliant on the mechanism of action of the antibiotic used. Therefore, the reviewer should have focused on studies closely related to aminoglycoside tolerance. Our study is not confusing or contradictory, as Lewis’ group also demonstrated that the tolerance of the *icd* mutant to gentamicin was significantly reduced while the tolerance of other TCA cycle mutant strains was increased in a different study (Shan et al., 2015). However, they did not delve into the metabolism of these mutant strains, as we did. We now mention this point in our manuscript (see page 14).

Apart from the confusing data, it is not clear what useful information may be obtained from the choice of the experimental system. The authors examine exponentially growing cells of *E. coli* for tolerance of aminoglycosides. The population at this stage of growth is highly susceptible to aminoglycosides, and only some rare persister cells can survive. However, the authors do not study persisters. A stationary population of *E. coli* is tolerant of aminoglycosides, and this is clinically relevant, but this is not the subject of the study.

Response: Respectfully, we must express our disagreement with the reviewer's comments. Our experimental system is meticulously organized and logically structured. Mutant strains such as *gltA*, *sucA*, and *nuoI* deletions exhibit increased tolerance to all aminoglycosides tested, with their fractions clearly increasing around the mid-exponential phase between 3-4 hours (refer to Figure 2B in our manuscript). This surge in tolerance is evident at the population level as well (as depicted in Figure 1A in our manuscript, where certain mutant strains demonstrate complete survival to streptomycin, with survival fractions nearing 1). Given the pronounced increase observed around the mid-exponential phase, we primarily characterize the metabolism of these cells during this growth phase.

It's essential to note that any investigation into antibiotic tolerance and/or resistance holds immense significance, regardless of the growth phase under scrutiny, as antibiotic tolerance/resistance poses a substantial healthcare challenge. Additionally, metabolic mutant strains do not necessarily entail severe fitness costs, as evidenced by Figure S2A published by the Lewis group (Manuse et al., 2021), a finding consistent with our study (see Figure 2A in our manuscript). This phenomenon could confer a survival advantage to bacterial cells, as they may acquire metabolic mutations to bolster their tolerance without incurring significant fitness costs. Furthermore, numerous studies suggest that bacterial cells may opt for the evolutionary pathway leading to increased tolerance before acquiring resistance mechanisms (Levin-Reisman et al., 2017; Santi et al., 2021). The presence of metabolic mutations in clinical *E. coli* pathogens has also been confirmed through the analysis of a large library of 7243 *E. coli* genomes from NCBI Pathogen Detection by Collin’s group (Lopatkin et al., 2021). Consequently, comprehending the tolerance mechanisms of metabolic mutations holds paramount importance.

References

Levin-Reisman I, Ronin I, Gefen O, Braniss I, Shoresh N, Balaban NQ. 2017. Antibiotic tolerance facilitates the evolution of resistance. *Science (1979)* 355:826–830. doi:10.1126/science.aaj2191

Lopatkin AJ, Bening SC, Manson AL, Stokes JM, Kohanski MA, Badran AH, Earl AM, Cheney NJ, Yang JH, Collins JJ. 2021. Clinically relevant mutations in core metabolic genes confer antibiotic resistance. *Science (1979)* 371. doi:10.1126/science.aba0862

Manuse S, Shan Y, Canas-Duarte SJ, Bakshi S, Sun WS, Mori H, Paulsson J, Lewis K. 2021. Bacterial persisters are a stochastically formed subpopulation of low-energy cells. *PLoS Biol* 19. doi:10.1371/journal.pbio.3001194

Mohiuddin Kabir M, Shimizu K. 2004. Metabolic regulation analysis of icd-gene knockout *Escherichia coli* based on 2D electrophoresis with MALDI-TOF mass spectrometry and enzyme activity measurements. *Appl Microbiol Biotechnol* 65:84–96. doi:10.1007/s00253-004-1627-1

Santi I, Manfredi P, Maffei E, Egli A, Jenal U. 2021. Evolution of Antibiotic Tolerance Shapes Resistance Development in Chronic *Pseudomonas aeruginosa* Infections. doi:10.1128/mBio.03482-20

Shan Y, Lazinski D, Rowe S, Camilli A, Lewis K. 2015. Genetic basis of persister tolerance to aminoglycosides in *Escherichia coli. mBio* 6. doi:10.1128/mBio.00078-15

Van den Bergh B, Schramke H, Michiels JE, Kimkes TEP, Radzikowski JL, Schimpf J, Vedelaar SR, Burschel S, Dewachter L, Lončar N, Schmidt A, Meijer T, Fauvart M, Friedrich T, Michiels J, Heinemann M. 2022. Mutations in respiratory complex I promote antibiotic persistence through alterations in intracellular acidity and protein synthesis. *Nat Commun* 13:546. doi:10.1038/s41467-022-28141-x

**Reviewer #2 (Public Review):**
Summary:This interesting study challenges a dogma regarding the link between bacterial metabolism decrease and tolerance to aminoglycosides (AG). The authors demonstrate that mutants well-known for being tolerant to AG, such as those of complexes I and II, are not so due to a decrease in the proton motive force (PMF) and thus antibiotic uptake, as previously reported in the literature.Strengths:This is a complete study. These results are surprising and are based on various read-outs, such as ATP levels, pH measurement, membrane potential, and the uptake of fluorophore-labeled gentamicin. Utilizing a proteomic approach, the authors show instead that in tolerant mutants, there is a decrease in the levels of proteins associated with ribosomes (targets of AG), causing tolerance.

Response: We sincerely appreciate the reviewer for taking the time to read our manuscript and offer valuable suggestions.

Weaknesses:The use of a single high concentration of aminoglycoside: my main comment on this study concerns the use of an AG concentration well above the MIC (50 µg/ml or 25 µg/ml for uptake experiments), which is 10 times higher than previously used concentrations (Kohanski, Taber) in study showing a link with PMF. This significant difference may explain the discrepancies in results. Indeed, a high concentration of AG can mask the effects of a metabolic disruption and lead to less specific uptake. However, this concentration highlights a second molecular level of tolerance. Adding experiments using lower concentrations (we propose 5 µg/ml to compare with the literature) would provide a more comprehensive understanding of AG tolerance mechanisms during a decrease in metabolism.Another suggestion would be to test iron limitation (using an iron chelator as DIP), which has been shown to induce AG tolerance. Can the authors demonstrate if this iron limitation leads to a decrease in ribosomal proteins? This experiment would validate their hypothesis in the case of a positive result. Otherwise, it would help distinguish two types of molecular mechanisms for AG tolerance during a metabolic disruption: (i) PMF and uptake at low concentrations, (ii) ribosomal proteins at high concentrations.

Response: While we acknowledge the intriguing possibility of exploring whether iron limitation results in a reduction of ribosomal proteins, we believe that this topic falls slightly outside the scope of our current study. This area warrants independent investigation since our current research did not specifically focus on iron-limited environments (LB medium is iron-rich, as referenced (Abdul-tehrani et al., 1999; Rodríguez-Rojas et al., 2015)). However, we fully concur with the notion that experimental outcomes may be contingent upon the concentration of aminoglycosides (AG). Hence, we repeated the critical experiments using a lower concentration of gentamicin (5 µg/mL), as suggested by the reviewer. Before delving into a discussion of these results, we wish to emphasize two key points. Firstly, the majority of our metabolic measurements, including ATP levels, redox activities, intracellular pH, and metabolomics, were conducted in mutant and wild-type cells in the absence of drugs. Our objective was to elucidate the impact of genetic perturbations of the TCA cycle on cell metabolism. Secondly, it's important to emphasize that our study does not invalidate the hypothesis that AG uptake is proton motive force (PMF)-dependent. We observed similar drug uptake across the strains tested, which is reasonable considering that their energy metabolism and PMF are not significantly altered compared to the wild type (at least we did not observe a consistent trend in their metabolic levels). Consequently, our study does not necessarily contradict with previous claims (53). We have now clarified this point in the manuscript (see pages 3 and 12).

When we employed a lower gentamicin concentration, we still noted a significant elevation in tolerance among the *gltA*, *sucA*, and *nuoI* mutant strains compared to the wild type. Also, it remained evident that the observed tolerance in the mutant strains cannot be ascribed to differences in drug uptake or impaired PMF, as the levels of drug uptake and the disruption of PMF by gentamicin (at lower concentrations) in the mutant strains were comparable to those of the wild type. Moreover, since our metabolic measurements and proteomics analyses failed to reveal any notable alterations in energy metabolism in these strains, the consistency in drug uptake levels across both mutant and wild-type strains, even at lower concentrations, further bolsters the validity of our findings obtained at higher gentamicin concentrations. The new results have been incorporated (see Figure 1 – figure supplement 1, Figure 4 – figure supplement 1, Figure 5 – figure supplement 2, Figure 5 – figure supplement 4) and discussed throughout the manuscript.

**Recommendations for the authors:**

**Reviewer #2 (Recommendations For The Authors):**
Line 120: Luria-Bertani (LB), used Lysogeny Broth.Line 180: "RSG dye can be reduced by bacterial reductases of PMF" to be reformulated.

Response: The suggested corrections have been incorporated into the manuscript.

References

Abdul-tehrani H, Hudson AJ, Chang Y, Timms AR, Hawkins C, Williams JM, Harrison PM, Guest JR, Andrews SC. 1999. Ferritin Mutants of *Escherichia coli* Are Iron Deficient and Growth Impaired, and fur Mutants are Iron Deficient, *Journal of Bacteriology*.

Rodríguez-Rojas A, Makarova O, Müller U, Rolff J. 2015. Cationic Peptides Facilitate Iron-induced Mutagenesis in Bacteria. *PLoS Genet* 11. doi:10.1371/journal.pgen.1005546

Taber Harry W, Mueller JP, Miller PF, Arrow AS. 1987. Bacterial Uptake of Aminoglycoside Antibiotics. *Microbiol Rev* 51:439–457. doi:10.1128/mr.51.4.439-457.1987